# ADASINE-LORA: ADAPTIVE FREQUENCY MODULATION FOR NONLINEAR LOW-RANK ADAPTATION

## ABSTRACT

Low-Rank Adaptation (LoRA) has emerged as an efficient fine-tuning paradigm for large models by injecting trainable low-rank updates into frozen weights. However, the inherent linearity and low-rank nature of LoRA restrict its capacity to capture complex nonlinear semantics. Recent work demonstrates that applying nonlinear functions such as sine to the low-rank component can significantly enhance its expressiveness. Despite this, these methods typically rely on a static frequency, failing to accommodate the input-dependent variations in optimal perturbation scale. In this paper, we propose AdaSine-LoRA, a novel framework that integrates adaptive frequency modulation into the sine-activated LoRA formulation. Instead of using a fixed global frequency, our method dynamically generates a frequency coefficient conditioned on the input, enabling input-aware control over the perturbation pattern. We analyze the relationship between frequency and the effective rank of the perturbed weight space, and empirically demonstrate that adaptive frequency leads to consistently improved performance across diverse tasks with minimal parameter overhead. Our approach provides a lightweight yet effective mechanism to enhance LoRA's expressivity by aligning perturbation dynamics with task-specific input structures. Our extensive experiments demonstrate that this design consistently outperforms LoRA and its variants across a wide range of downstream tasks, including large language model fine-tuning and visual instruction tuning.

## 1 INTRODUCTION

Large-scale pre-trained models have demonstrated remarkable generalization capabilities across diverse domains, including natural language processing (Brown et al., 2020; Zhao et al., 2023), computer vision (He et al., 2022; Dosovitskiy et al., 2021), and multi-modal learning (Li et al., 2022; Liu et al., 2023a). These models, however, typically require adaptation to downstream tasks, where fully fine-tuning all parameters is often computationally prohibitive and memory-intensive. To mitigate this, parameter-efficient fine-tuning (PEFT) strategies have emerged as practical alternatives, enabling task adaptation by training only a small fraction of the model's parameters (Hu et al., 2022; Houlsby et al., 2019).

Among these methods, LoRA (Hu et al., 2022) has gained widespread popularity due to its balance of simplicity, compatibility, and effectiveness. LoRA introduces trainable low-rank matrices into each layer of a frozen backbone, effectively approximating the full-parameter updates while dramatically reducing storage and compute requirements. Despite its success, LoRA's linear structure and fixed rank impose significant expressiveness constraints. Specifically, its ability to capture complex, non-linear feature transformations is inherently limited, especially when adapting to tasks involving high-dimensional, non-linear semantics (Kopiczko et al., 2024; Lialin et al., 2023).

To address these limitations, a growing body of work has explored enhancing LoRA's structural expressiveness. A particularly promising line of research introduces non-linear transformations into the low-rank pathway. Among various candidates, the sine function has emerged as a theoretically and empirically effective choice (Li et al., 2024), owing to its smoothness, periodicity, and ability to approximate high-rank behavior without additional parameters. These properties allow sine-based LoRA variants to significantly improve performance while retaining LoRA's parameter-efficiency. However, existing sine-enhanced LoRA methods typically rely on a fixed frequency

Figure 1: Architectural comparison of LoRA-based adaptation methods: (a) Standard LoRA, (b) SineLoRA with fixed-frequency modulation, and (c) our proposed AdaSine-LoRA with input-conditioned modulation.

parameter, shared across all tokens and tasks. This static design fails to account for the diverse perturbation magnitudes required across different input instances, leading to suboptimal performance in practice. Moreover, the optimal frequency often exhibits strong correlation with the underlying data distribution and knowledge structure. As a result, an improperly chosen fixed frequency can induce over-parameterization, compromising both generalization and training efficiency.

In this work, we propose **AdaSine-LoRA**. Different from the existing methods, this is a new enhancement of LoRA, which introduces an input conditional frequency modulation mechanism. As illustrated in Figure1(a), standard LoRA injects a learnable low-rank residual into the frozen pretrained weight matrix using two projection matrices. While this design enables parameter-efficient fine-tuning, the low-rank constraint inherently limits the representational capacity of the update. To address this limitation, recent work introduces nonlinear transformations into the low-rank pathway, as shown in Figure1(b). In particular, applying sine activation to the low-rank component has been shown to increase the functional rank of the update, effectively enhancing model expressiveness while adding only minimal parameters. Building upon this insight, our proposed AdaSine-LoRA (Figure1(c)), Rather than relying on a global frequency hyperparameter, our method learns a lightweight function that maps input features to a frequency coefficient, dynamically modulating the scale of the sine transformation. This enables token-wise control over the non-linear update strength, allowing the model to tailor its expressive capacity to the complexity of each input.

We provide theoretical analysis linking the frequency parameter to the effective rank of the perturbed update, and show that adaptive modulation allows the model to maintain expressiveness with introducing only negligible additional parameters count. We also empirically validate our approach on a wide range of language and vision tasks, demonstrating consistent improvements over standard LoRA and its non-linear variants.

Our contributions are summarized as follows:

- We propose AdaSine-LoRA, which introduces input-aware frequency modulation into the sine-transformed LoRA updates, enabling adaptive and expressive fine-tuning with almost no additional parameters.
- We propose an efficient implementation of the nonlinear LoRA computation, introducing negligible additional fine-tuning cost while significantly enhancing model expressiveness.
- We provide intuitive analyses of the role of frequency in shaping the effective rank of nonlinear low-rank updates, complemented by extensive experiments across NLP and vision tasks showing clear improvements over prior LoRA variants.

## 2 BACKGROUND AND MOTIVATION

### 2.1 LORA BASICS

LoRA has become a standard technique for parameter-efficient fine-tuning, wherein the pre-trained weights $W_0$ are frozen and a learnable low-rank residual $\Delta W$ is inserted at each layer of a pretrained model. Specifically, for each layer, LoRA represents the residual using two consecutive low-rank matrices $B \in \mathbb{R}^{d \times r}$, $A \in \mathbb{R}^{r \times k}$, where $d$ denotes the output dimension, $k$ the input

dimension, and $r$ the rank of the decomposition, such that:

$$\Delta W = BA, \tag{1}$$

with $r \ll \min(d, k)$. This design yields significant parameter savings.

However, the expressive capacity of $\Delta W$ is inherently constrained by its rank. Let $\Delta W^* \in \mathbb{R}^{d \times k}$ denote the ideal residual matrix that perfectly aligns the pre-trained weights with the fine-tuned objective. The goal of adaptation is to approximate $\Delta W^*$ using the restricted structure $\Delta W = BA$. Prior analysis by Hu et al. (Hu et al., 2022) has shown that LoRA can be regarded as a general approximation to full fine-tuning, and its expressive capacity approaches that of full adaptation as $r$ grows. This insight highlights both the strength and the limitation of LoRA: while higher rank strictly improves approximation quality in principle, practical deployments constrain $r$ to remain small for efficiency, thereby limiting the attainable representational power. This motivates our focus on mechanisms that can effectively increase the attainable rank of LoRA updates without incurring prohibitive parameter cost.

## 2.2 Non-linear LoRA

To overcome this limitation, several recent works have proposed enhancing LoRA by incorporating non-linear transformations (Jiang et al., 2024; Valipour et al., 2022). Among these, the sine-based method (Ji et al.) stands out for its use of a smooth, periodic, and bounded function to increase the effective rank of the transformation. Specifically, it replaces the standard residual with:

$$\Delta W = \frac{1}{g} \cdot \sin(\omega \cdot BA), \tag{2}$$

where $\omega \in \mathbb{R}$ is a fixed frequency hyperparameter, and $g$ is a scaling factor for stability.

This method has been theoretically shown to enhance representational capacity while maintaining the parameter-efficiency of LoRA. Among various candidate non-linearities, the sine function offers the most favorable trade-off in terms of expressiveness and stability. However, a key limitation of this formulation is that the frequency parameter $\omega$ is treated as a manually chosen scalar hyperparameter, this fixed design limits the adaptability of the model across tasks and inputs.

## 2.3 Observations

**Observation 1: Frequency positively correlates with effective rank.** Figure 2(a)-(b) shows that the average rank initially grows linearly with increasing frequency $\omega$, but eventually saturates. Here, $B \in \mathbb{R}^{d \times r}$ and $A \in \mathbb{R}^{r \times d}$ are initialized via Kaiming uniform distribution. To further examine this effect, we conducted a controlled initialization study (Figure 2(c)) using a low-rank matrix $BA \in \mathbb{R}^{512 \times 512}$, with $B \in \mathbb{R}^{512 \times 2}$ and $A \in \mathbb{R}^{2 \times 512}$. Applying $\sin(\omega BA)$ at discrete frequencies $\omega \in \{5, 50, 100, 400\}$, we visualized row-wise cosine similarities. The heatmaps reveal that larger $\omega$ reduces redundancy and diversifies row directions, leading to higher effective rank. While moderate $\omega$ increases improve representation, excessively high values may push the structure toward an overly high-rank space, resulting in over-parameterization and degraded generalization.

**Observation 2: Model capability does not increase monotonically with matrix rank.** Although higher frequency usually leads to higher rank, this does not always yield better task performance. As illustrated in Figure 3, we report the accuracy across varying rank settings of LoRA on four representative commonsense reasoning datasets: PIQA, SIQA, WinoGrande, and OBQA. We observe a non-monotonic trend, where accuracy peaks at an intermediate rank and then declines, likely due to overfitting. Combined with Observation 1, this suggests that larger $\omega$ may not always be desirable in practice, and an optimal frequency range likely exists.

**Observation 3: The optimal frequency varies significantly across datasets.** In Figure 4, we plot model performance under varying fixed $\omega$ values across several datasets. Models are trained and evaluated on GSM8K, HumanEval (fine-tuned on Code Alpaca 20K), and HellaSwag. Each dataset exhibits a different peak frequency, indicating that the optimal $\omega$ is sensitive to input distribution. Moreover, frequency is not just a secondary factor, but a critical component of the adaptation mechanism.

Taken together, these observations reveal the limitations of a fixed-frequency design. This motivates our proposal to learn a token-wise input-conditioned frequency function $\omega(x)$, allowing the

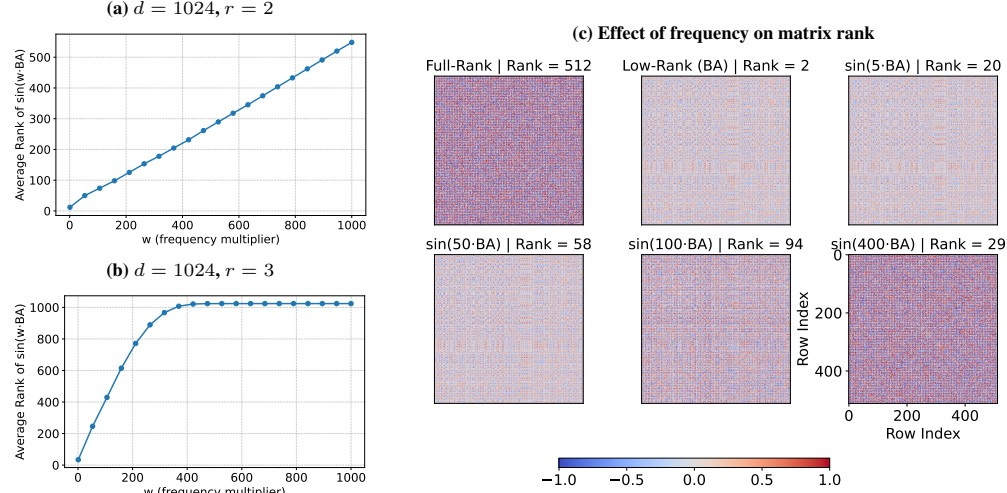

Figure 2: **Effect of Frequency Multiplier on Matrix Rank and Structure.** (a)-(b) Relationship between the $\omega$ and the average rank of the transformed matrix $\sin(\omega \cdot BA)$ for different matrix parameter settings $(d, r)$, where $d$ is the matrix dimension and $r$ is the rank of the initial low-rank matrix $BA \in \mathbb{R}^{d \times d}$. (c) Cosine similarity heatmaps of initialized matrices $\sin(\omega BA)$ under discrete frequencies $\omega$. The heatmaps visualize pairwise row-wise similarity using $1 - |\cos(x_i, x_j)|$ with sign preservation, where darker regions indicate lower similarity and thus higher effective rank.

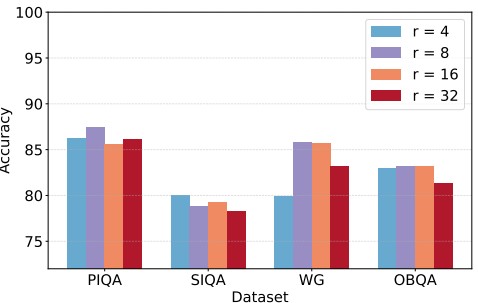

Figure 3: Accuracy of LLaMA3-8B on Commonsense Reasoning Datasets with Different LoRA Ranks $r$. Each group of bars corresponds to a dataset (PIQA, SIQA, Wino-Grande, and OBQA), and each bar within a group represents a different LoRA rank.

Figure 4: Cross-Dataset Frequency Sensitivity. Each line shows a LLaMA3-8B model fine-tuned with a fixed sine frequency $\omega$. Performance is centered by subtracting the mean per curve, with red dots marking the best $\omega$ for each dataset.

model to dynamically modulate the non-linearity and thereby achieve fine-grained, data-dependent adaptation.

## 3 METHOD

In this section, we first analyze how applying a sine nonlinearity to a low-rank update matrix affects its effective rank, and show that the modulation frequency $\omega$ directly controls the strength of rank expansion. This theoretical perspective naturally motivates making $\omega$ input-dependent, leading to our AdaSine-LoRA formulation.

**Rank Transition Mechanism.** Given a low-rank update matrix $M = BA \in \mathbb{R}^{d \times k}$ with $\text{rank}(M) = r$, the sine-activated update admits an element-wise Taylor expansion:

$$\sin(\omega M) = \omega M - \frac{\omega^3}{3!} M^{\odot 3} + O(\omega^5),$$

where $M^{\odot 3}$ denotes the element-wise cubic power. This decomposition reveals that $\sin(\omega M)$ consists of a rank-$r$ term $\omega M$ plus a nonlinear perturbation whose magnitude scales as $O(\omega^3)$. Under standard randomness and genericity assumptions on the low-rank factors $A$ and $B$, the higher-order component $M^{\odot 3}$ (and subsequent odd-order terms) does not lie in the original rank-$r$ subspace, thereby introducing additional independent directions in the row and column spaces.

As $\omega$ increases, the perturbation strength grows, progressively breaking the low-rank structure and yielding a monotonic rise in the *effective* numerical rank until saturation near full rank. This perspective suggests that the frequency $\omega$ is a direct knob controlling the extent of rank expansion: small $\omega$ keeps the update close to the original low-rank structure, while larger $\omega$ induces stronger nonlinear perturbations and higher effective rank. Consequently, for each input, an ideally chosen $\omega$ should induce the most beneficial perturbation magnitude.

For a more detailed mathematical treatment of the $\omega$–rank relationship and the input-conditioned frequency selection, refer to the analysis provided in the Theoretical Framework section of the Appendix (Section A.2).

**From Fixed to Input-Conditioned Frequency.**    The above analysis indicates that the optimal frequency $\omega$ is not universal, but should depend on how much rank expansion is beneficial for a given input. Intuitively, different tokens may require different levels of representational flexibility, and a fixed-frequency transformation fails to accommodate this diversity. To align the perturbation scale with input-dependent needs, we introduce an input-conditioned frequency function $\omega(x)$, enabling each token to adaptively modulate the strength of the non-linearity based on its specific characteristics.

Concretely, for each input token $x \in \mathbb{R}^k$, we predict a scalar frequency via a trainable linear transformation:

$$\omega(x) = W_\omega x, \quad W_\omega \in \mathbb{R}^{1 \times k}. \tag{3}$$

This design is computationally lightweight and introduces negligible parameter overhead compared to standard LoRA. However, directly using the raw output of $\omega(x)$ may result in unstable training dynamics, especially during the early stages when weights are randomly initialized. Large frequency values amplify the oscillation of the sine function and can lead to gradient explosion.

**Stable Frequency Normalization.**    To ensure numerical stability while retaining the rank-expansion benefits of sine activation, we introduce a sigmoid-based normalization to bound the predicted frequency within a safe and effective range:

$$\tilde{\omega}(x) = \omega_0 \cdot \sigma(\omega(x)), \tag{4}$$

where $\omega_0$ is a global maximum frequency hyperparameter and $\sigma(\cdot)$ is the sigmoid function. This normalization constrains $\tilde{\omega}(x) \in (0, \omega_0)$, thereby guaranteeing smooth gradients and stable optimization behavior across training iterations. In light of the rank transition analysis, $\omega_0$ sets an upper bound on the maximal rank expansion, while $\sigma(W_\omega x)$ allows each token to select an appropriate perturbation scale within this range.

**From Matrix-Centric to Function-Centric Formulation.**    A naive formulation would be to construct the full transformed matrix first,

$$\Delta W = \frac{1}{g} \cdot \sin(\tilde{\omega}(x) \cdot BA), \tag{5}$$

where $g$ is a scaling factor that controls the magnitude of the update. Applying it to an input gives

$$\Delta W x = \frac{1}{g} \cdot \sin(\tilde{\omega}(x) \cdot BA) \, x. \tag{6}$$

However, this formulation requires instantiating a distinct matrix for each input token due to the input-dependent frequency $\tilde{\omega}(x)$, resulting in a large activation tensor of shape $(b, l, d, k)$, where $b$ is the batch size and $l$ is the sequence length. The memory usage in this case scales as $\mathcal{O}(bldk)$, which becomes prohibitive in large-scale models and long-sequence tasks, especially when $k$ (input dimension) is large. This makes the naive approach inefficient for practical applications.

Rather than explicitly constructing $\Delta W$ and then applying it to $x$, we instead model its *action* on the input, i.e., directly approximate $\Delta W x$. This reformulation can be seen as moving from a matrix-centric approximation view ($\Delta W \approx BA$) to a function-centric one ($\Delta W x \approx f(BAx)$). Since the ultimate effect of $\Delta W$ is only realized when multiplied with $x$, working with the functional form is both mathematically near-equivalent and computationally more efficient. Concretely, we compute:

$$\Delta W x = \frac{1}{g} \cdot \sin\left(\tilde{\omega}(x) \cdot BAx\right). \tag{7}$$

This avoids constructing per-token full matrices and instead directly operates in a token-wise manner. In practice, we first obtain the low-rank projection $Ax \in \mathbb{R}^r$, then lift it via $BAx \in \mathbb{R}^d$, and finally apply the sine transformation. The largest activation tensor here has shape $(b, l, d)$, identical to standard LoRA, thereby retaining its efficiency. This design yields substantial memory savings—approximately a factor of $\mathcal{O}(k)$—while preserving the expressive benefits of sine-based transformations. Crucially, it allows the model to approximate the functional behavior of the original residual through token-specific nonlinear perturbations, providing greater modeling flexibility than a static matrix update.

**Final Formulation.** Putting all components together, our final adaptive update rule is:

$$\Delta W x = \frac{1}{g} \cdot \sin\left(\omega_0 \cdot \sigma(W_\omega x) \cdot BAx\right). \tag{8}$$

This formulation supports input-aware adaptation, avoids the high memory cost of full matrix instantiation, and introduces bounded non-linearity for training stability. It maintains the parameter- and compute-efficiency of standard LoRA, while significantly enhancing its expressiveness, adaptability, and robustness.

## 4 EXPERIMENTS

In this section, we present experiments demonstrating the effectiveness of AdaSine-LoRA across diverse tasks spanning both language and vision–language models. We first fine-tune LLaMA 3 on commonsense reasoning benchmarks to compare AdaSine-LoRA with several PEFT baselines. We then assess its applicability to generative tasks using standard NLG benchmarks. Finally, we extend the evaluation to multimodal settings by comparing AdaSine-LoRA with other LoRA variants on image–text understanding tasks using LLaVA-1.5-7B.

### 4.1 LARGE LANGUAGE MODELS

To comprehensively validate the effectiveness of AdaSine-LoRA on large language models (LLMs), we conduct extensive experiments by fine-tuning LLaMA 3-8B on both commonsense reasoning and NLG tasks.

**Dataset.** For commonsense reasoning, we utilize eight benchmark datasets, including BoolQ(Clark et al., 2019), PIQA(Bisk et al., 2019), SIQA(Sap et al., 2019), HellaSwag (HS)(Zellers et al., 2019), WinoGrande (WG)(Sakaguchi et al., 2021), ARC-c, ARC-e(Clark et al., 2018), and OBQA(Mihaylov et al., 2018). All training splits from these datasets are merged to form a unified training set, while evaluations are performed separately on the official test sets of each benchmark. For NLG tasks, we fine-tune the model on the MetaMath-40K(Yu et al., 2023) dataset and evaluate it on both GSM8K (Cobbe et al., 2021) and Math(Yu et al., 2023) benchmarks. We also perform fine-tuning on CodeAlpaca(Chaudhary, 2023), followed by evaluations on HumanEval(Chen et al., 2021) and MBPP. Additionally, we fine-tune on the XSum training set and evaluate on its official validation set.

**Experiment Setup.** To evaluate the effectiveness of AdaSine-LoRA on LLaMA 3-8B, we follow the standard LoRA design and apply our method to the feed-forward network (FFN) modules of each transformer layer. Following SineLoRA(Ji et al.), we set $g = \sqrt{n}$, where $n$ is the number of rows of the weight matrix. We compare the performance of vanilla LoRA, its variants DoRA and Sine-LoRA, and our proposed AdaSine-LoRA under different rank settings $r = 8, 16, 32$. The maximum frequency $\omega_0$ in AdaSine-LoRA is set to 400. Additional training configurations and hyperparameters are detailed in the Appendix.

Table 1: Performance and parameter count of the LLaMA 3-8B model fine-tuned using AdaSine-LoRA and other LoRA variants across varying $r$ settings on the commonsense reasoning benchmark.

| Method | Params | BoolQ | PIQA | SIQA | HS | WG | ARC-e | ARC-c | OBQA | Avg. | $\Delta$ vs. LoRA |
|---|---|---|---|---|---|---|---|---|---|---|---|
| LoRA$_{r=8}$ | 14.2M | 89.66 | 86.24 | 77.89 | 92.97 | 85.40 | 87.13 | 74.94 | 83.60 | 84.73 | |
| Sine LoRA$_{r=8}$ | 14.2M | 89.94 | 86.94 | 81.27 | 93.75 | 85.71 | 85.19 | 79.40 | 87.00 | 86.15 | 1.42↑ |
| Dora$_{r=8}$ | 15.2M | 90.00 | **89.99** | 82.86 | 95.50 | 81.06 | 89.77 | 82.83 | 89.40 | 87.67 | 2.94↑ |
| Sine DoRA$_{r=8}$ | 15.2M | **90.34** | 87.05 | 82.09 | 94.91 | 82.40 | 89.95 | 82.75 | 89.80 | 87.41 | 2.68↑ |
| AdaSine-LoRA$_{r=8}$ | 14.9M | 90.15 | 88.63 | **82.24** | **95.54** | **87.85** | **93.65** | **83.61** | **90.20** | **88.98** | 4.25↑ |
| LoRA$_{r=16}$ | 28.3M | 89.79 | 85.96 | 77.18 | 92.53 | 85.87 | 88.01 | 75.11 | 83.00 | 84.68 | |
| Sine LoRA$_{r=16}$ | 28.3M | 90.24 | 87.32 | 81.83 | 94.33 | 87.69 | 90.65 | 81.55 | 87.80 | 87.68 | 3.00↑ |
| Dora$_{r=16}$ | 29.4M | 90.31 | 89.17 | 82.40 | **95.74** | 86.82 | 88.18 | 82.92 | 89.00 | 88.07 | 3.39↑ |
| Sine DoRA$_{r=16}$ | 29.4M | 90.46 | 88.08 | 82.54 | 95.40 | 87.61 | 87.65 | 82.58 | 89.80 | 88.02 | 3.34↑ |
| AdaSine-LoRA$_{r=16}$ | 29.0M | **90.52** | **89.45** | **83.11** | 95.58 | **88.40** | 92.77 | 83.09 | **90.60** | **89.19** | 4.51↑ |
| LoRA$_{r=32}$ | 56.6M | 90.06 | 87.98 | 77.89 | 93.85 | 83.03 | 89.24 | 77.51 | 85.80 | 85.67 | |
| Sine LoRA$_{r=32}$ | 56.6M | 90.70 | 87.81 | 82.09 | 94.57 | **86.82** | 88.89 | 81.55 | 87.60 | 87.50 | 1.83↑ |
| Dora$_{r=32}$ | 57.7M | 90.58 | 89.45 | **83.16** | 95.64 | 84.53 | 92.24 | 82.75 | 89.40 | 88.47 | 2.80↑ |
| Sine DoRA$_{r=32}$ | 57.7M | 90.00 | 89.72 | 82.09 | 95.16 | 85.40 | **93.65** | 82.75 | 89.40 | 88.52 | 2.85↑ |
| AdaSine-LoRA$_{r=32}$ | 57.3M | **90.80** | **90.64** | 83.06 | **95.83** | 86.66 | 93.12 | **84.46** | 89.80 | **89.30** | 3.63↑ |

**Main Results.** The experimental results of AdaSine-LoRA and competing baselines are summarized in Table 1 and Table 2. Across all benchmark datasets, AdaSine-LoRA consistently outperforms vanilla LoRA and its variants on both commonsense reasoning and NLG tasks. This superior performance can be attributed to the adaptive frequency modulation mechanism, which enables more flexible exploration of nonlinear function space while retaining controllable representational capacity.

Furthermore, the additional parameter cost introduced by the adaptive frequency-modulated linear layers is minimal, making AdaSine-LoRA a lightweight and scalable fine-tuning strategy for large-scale models. Notably, this overhead grows sublinearly compared to the parameter increase incurred by raising the rank $r$, offering a more cost-effective trade-off between performance and parameter count. Notably, AdaSine-LoRA exhibits clear advantages on complex NLG tasks such as code generation and mathematical reasoning, demonstrating strong robustness and the ability to accommodate tasks that demand higher representational expressivity.

## 4.2 VISUAL INSTRUCTION TUNING

To further investigate the scalability of AdaSine-LoRA, we extend our experiments to larger model settings by applying visual instruction tuning on VLMs. To ensure a fair comparison, we strictly follow the DoRA training configurations, i.e. adopting the same LoRA settings as used in its original implementation. Our experiments are carried out on LLaVA-1.5-7B(Liu et al., 2023a), which integrates the Vicuna-1.5-7B language model(Peng et al., 2023) and the CLIP ViT-L/336px vision encoder(Radford et al., 2021). The training corpus comprises a mixture of datasets, including VQA datasets(Schwenk et al., 2022), benchmarks(Mishra et al., 2019), region-level VQA datasets(Mao et al., 2016), visual dialogue data(Liu et al., 2023a), and standard language-only conversations. We follow the data filtering and prompt construction strategies introduced in(Liu et al., 2023a).

Evaluation is conducted on seven vision-language benchmarks: VQAv2(Goyal et al., 2017), GQA(Hudson & Manning, 2019), VizWiz(Gurari et al., 2018), SQA(Lu et al., 2022), VQAT(Singh et al., 2019), POPE(Li et al., 2023), and MMBench(Liu et al., 2023b). As shown in Table 3, LoRA already outperforms full fine-tuning (FT) in terms of average accuracy, potentially due to FT's tendency to overfit in certain scenarios. In such cases, AdaSine-LoRA further improves performance by leveraging its adaptive frequency modulation mechanism, which effectively mitigates overfitting and enhances generalization. Across most benchmarks, AdaSine-LoRA consistently surpasses all LoRA variants and FT, achieving an average improvement of **+0.4** over **DoRA** and **+1.5** over **FT**. Meanwhile, the number of trainable parameters of AdaSine-LoRA maintains a similar level to LoRA and DoRA.

Table 2: Performance and parameter count of the LLaMA 3-8B model fine-tuned using AdaSine-LoRA and other LoRA variants across varying $r$ settings on natural language generation benchmarks.

| Method | r | Params | GSM8K | MATH | HumanEval | MBPP | Xsum rouge1 | rougeLsum | Avg. |
|--------|---|--------|-------|------|-----------|------|--------|-----------|------|
| LoRA | | 14.2M | 71.87 | 23.42 | 39.02 | 32.60 | 41.31 | 33.34 | 40.26 |
| Sine LoRA | 8 | 14.2M | 72.23 | 23.68 | 39.63 | 37.80 | 42.45 | 35.23 | 41.83 |
| DoRA | | 15.2M | **72.86** | 23.86 | 44.51 | 39.60 | 41.87 | 34.65 | 42.89 |
| AdaSine-LoRA | | 14.9M | 72.40 | **23.96** | **45.12** | **42.60** | **44.29** | **36.91** | **43.71** |
| LoRA | | 28.3M | 72.02 | 23.50 | 42.68 | 33.60 | 43.50 | 35.95 | 41.88 |
| Sine LoRA | 16 | 28.3M | 72.38 | 23.94 | 39.63 | 41.20 | 44.22 | 36.43 | 42.97 |
| DoRA | | 29.4M | 72.02 | 24.12 | 42.68 | 41.40 | **44.37** | **37.12** | 43.65 |
| AdaSine-LoRA | | 29.0M | **72.61** | **24.46** | **43.90** | **43.40** | 44.26 | 36.81 | **44.24** |
| LoRA | | 56.6M | 72.40 | 24.52 | 42.68 | 39.80 | 43.21 | 35.99 | 43.10 |
| Sine LoRA | 32 | 56.6M | 72.91 | 24.70 | 41.46 | 41.80 | 43.87 | 36.73 | 43.58 |
| DoRA | | 57.7M | **74.00** | 24.30 | 43.29 | 42.60 | 43.14 | 36.13 | 43.91 |
| AdaSine-LoRA | | 57.3M | 73.37 | **24.88** | **43.90** | **45.40** | **44.64** | **36.95** | **44.86** |

Table 3: Visual instruction tuning evaluation result of AdaSine-LoRA, DoRA, LoRA, and FT for LLaVA-1.5-7B on a wide range of 7 vision-language tasks.

| Method | %Param | VQAv2 | GQA | VisWiz | SQA | VQAT | POPE | MMBench | Avg |
|--------|--------|-------|-----|--------|-----|------|------|---------|-----|
| FT | 100 | 78.5 | 61.9 | 50.0 | 66.8 | **58.2** | 85.9 | 64.3 | 66.5 |
| LoRA | 4.61 | **79.1** | 62.9 | 47.8 | 68.4 | **58.2** | 86.4 | 66.1 | 66.9 |
| Sine LoRA | 4.61 | **79.1** | **63.4** | 51.4 | 69.4 | 57.2 | 87.4 | 66.1 | 67.7 |
| DoRA | 4.63 | 78.6 | 62.9 | 52.2 | **69.9** | 57.0 | 87.2 | 66.1 | 67.6 |
| Ours | 4.63 | 78.9 | 63.1 | **52.9** | 68.0 | 58.1 | **87.9** | **67.1** | **68.0** |

## 4.3 ANALYSIS

**Effect of the upper frequency bound $\omega_0$ on performance.** We conduct a series of experiments across eight commonsense reasoning datasets. Figure 5 reports the results on three representative datasets—HellaSwag, SIQA, and WinoGrande—along with the overall average performance. Horizontal dashed lines represent SineLoRA with fixed frequency $\omega = 200$, serving as a comparison baseline. The results demonstrate that AdaSine-LoRA is not particularly sensitive to the choice of $\omega_0$, as it achieves consistently strong performance across all tested values. This indicates that the effective range of $\omega_0$ is broad. Additionally, across most settings, the adaptive frequency modulation strategy consistently outperforms its fixed-frequency counterparts, suggesting that our method is generally more effective under a wide range of parameter configurations.

**Effect of different sine parameterizations.** Table 4 presents an ablation study comparing three ways of integrating the sine nonlinearity into the low-rank update: (i) applying sine after the low-rank perturbation ($\sin(w_0 BA) \cdot x$), (ii) applying sine directly to the transformed input ($\sin(w_0 BAx)$), and (iii) our adaptive frequency–modulated formulation ($\sin(\omega_0 \sigma(W_\omega x)BAx)$). Moving the sine operation inside the transformation already improves both memory efficiency and inference speed while preserving accuracy. Introducing input-dependent frequency further enhances model expressiveness, producing consistent gains under both $r = 16$ and $r = 32$. These results confirm that adaptive modulation provides a more flexible perturbation pattern and leads to the best overall performance–efficiency trade-off. Specifically, the results in Table show that "Mem. (GB)" refers to the training memory in gigabytes, while "Inf. Speed (it/s)" refers to the inference speed in iterations per second. These values were measured during model evaluation. The abbreviation "CR(Avg.)" in the table represents the average performance across all tasks in the "Commonsense

Table 4: Ablation of different sine parameterizations in LoRA. We compare applying sine outside the low-rank update, inside the transformed input, and our adaptive frequency modulation.

| Method | r | CR(Avg.) | GSM8K | MATH | Mem. (GB) | Inf. Speed (it/s) |
|---|---|---|---|---|---|---|
| $\sin(w_0 BA) \cdot x$ | 16 | 87.68 | 72.38 | 23.94 | 72.0 | 8.53 |
| $\sin(w_0 BAx)$ | 16 | 87.75 | 71.87 | 23.96 | **55.3** | **11.17** |
| $\sin(\omega_0 \sigma(W_\omega x)BAx)$ | 16 | **89.19** | **72.61** | **24.46** | 57.8 | 9.87 |
| $\sin(w_0 BA) \cdot x$ | 32 | 87.50 | 72.91 | 24.70 | 72.2 | 8.50 |
| $\sin(w_0 BAx)$ | 32 | 88.05 | 73.09 | 24.30 | **55.6** | **10.23** |
| $\sin(\omega_0 \sigma(W_\omega x)BAx)$ | 32 | **89.30** | **73.37** | **24.88** | 58.0 | 9.05 |

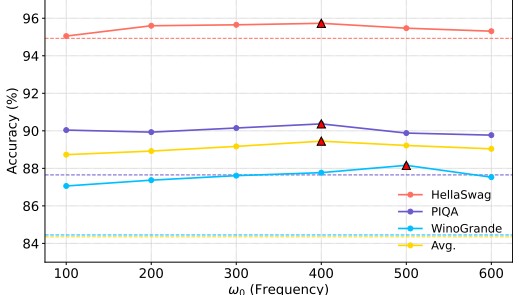
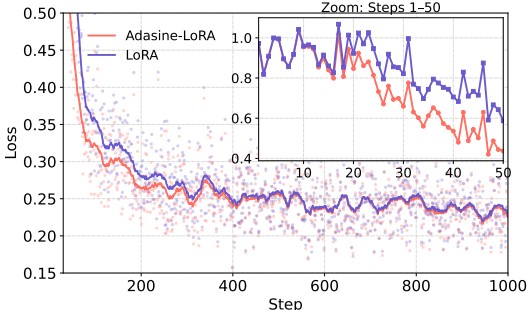

Figure 5: **Ablation study on frequency modulation with AdaSine-LoRA (rank = 16).** Performance is reported under varying frequency upper bounds $\omega_0$, with SineLoRA ($\omega = 200$) included as a baseline. For each dataset, the peak accuracy is marked with a red triangle.

Figure 6: **Loss curves on MetaMath (rank = 8).** The main plot shows training loss curves smoothed by a Savitzky–Golay filter (poly-order=3), with semi-transparent scatter points indicating raw data; the inset displays a zoomed-in view of steps 1–50.

Reasoning" dataset, further highlighting the balance between memory efficiency and model performance. The Inference Speed reported in the table is measured on the SIQA dataset.

**Training Convergence Analysis.** Experimental results show that AdaSine-LoRA achieves faster convergence with fewer training steps while enhancing task performance. Its nonlinear design with adaptive frequency modulation expands the representational space, enabling smoother weight updates and lower gradient variance. As shown in Figure 6, AdaSine-LoRA consistently reduces loss more quickly than standard LoRA, making it especially effective for large-scale models where stability and efficiency are critical.

**Efficiency Analysis.** Figure 7 compares the memory and time efficiency of different adaptation methods. As shown in subfigure (a), SineLoRA and SineDoRA incur substantially higher memory usage than LoRA, consistent with their design of applying sine activation after full-rank projection, which leads to larger intermediate tensors. In contrast, AdaSine-LoRA computes the low-rank projection $BAx$ before applying the sine function, effectively avoiding high-dimensional intermediate representations. This design enables AdaSine-LoRA to retain memory efficiency close to LoRA, while offering stronger expressiveness.

Subfigure (b) compares the actual training time of each method on the XSum dataset. The runtime is measured as the wall-clock time (in hours) required to complete one epoch on a single GPU, with a batch size of 2 and gradient accumulation of 16. AdaSine-LoRA again achieves a favorable result: it is only marginally slower than LoRA, but substantially faster than SineLoRA and DoRA. This efficiency gain can be attributed to the careful control of matrix sizes in our formulation. Since the dominant cost in large-scale training comes from matrix multiplications involving the weight update path, increasing the dimensionality of intermediate tensors—especially in methods like SineLoRA which propagate large $(d \times k)$ tensors per token—can dramatically slow down training. By operating directly on compact representations (i.e., projecting to $r$-dim space and computing $BAx$), and applying sine only after dimensionality has been reduced, AdaSine-LoRA avoids unnecessary

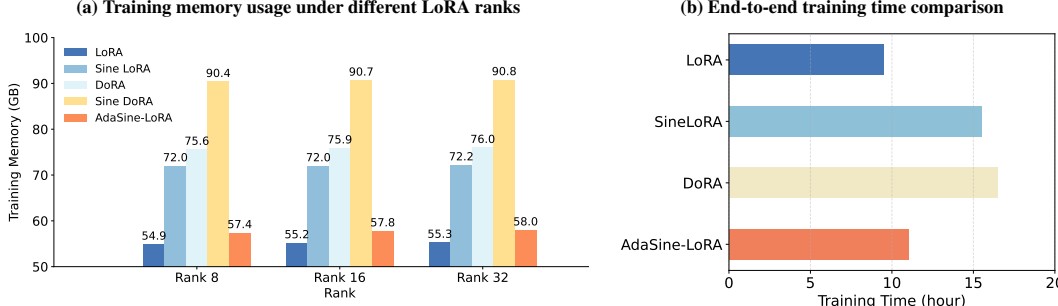

Figure 7: **Comparison of memory and training time across LoRA variants.** (a) Training memory consumption (in GB) of five methods under different LoRA ranks ($r = 8, 16, 32$), measured on the LLaMA3-8B model. (b) End-to-end training time (in hours) for four representative methods on the XSum summarization dataset. Each bar reports the time to train for one epoch using a batch size of 2 and gradient accumulation of 16 on a single GPU.

tensor expansion while still introducing non-linearity. This architectural discipline ensures that expressiveness is enhanced without compromising training scalability.

## 5 RELATED WORK

**Parameter-Efficient Fine-Tuning.** Parameter-efficient fine-tuning (PEFT) reduces computation and memory by updating only a small subset of parameters. Existing methods fall into three main categories: adapter-based, prompt-based, and reparameterization-based techniques. Adapter-based approaches (Houlsby et al., 2019; He et al., 2021; Karimi Mahabadi et al., 2021) insert lightweight modules into frozen backbones for efficient adaptation. Prompt-based methods such as Prefix Tuning (Li & Liang, 2021), Prompt Tuning (Lester et al., 2021), and P-Tuning v2 (Liu et al., 2022) optimize task-specific input tokens but often depend heavily on initialization. Reparameterization-based methods, typified by LoRA (Hu et al., 2022), restrict updates to a low-rank subspace without modifying the backbone or adding inference latency, with extensions like QLoRA (Dettmers et al., 2023) incorporating quantization-aware tuning.

**LoRA and Low-Rank Adaptation Improvements.** LoRA approximates fine-tuning updates using a pair of low-rank matrices, reducing memory cost but constraining expressiveness due to its fixed rank. Adaptive approaches such as IncreLoRA (Zhang et al., 2023) and DyLoRA (Valipour et al., 2022) address this by dynamically adjusting rank during training. Structural refinements like DoRA (Liu et al., 2024) decouple magnitude and direction, while routing-based methods including HydraLoRA (Tian et al., 2024) and AutoLoRA (Zhang et al., 2024b) introduce expert selection. Initialization techniques have also advanced: PiSSA (Meng et al., 2024) leverages SVD, whereas MiLoRA (Zhang et al., 2024a) and OLoRA (Büyükakyüz, 2024) use alternative spectral or QR-based forms. Other variants such as EVA (Paischer et al., 2024) and LoRA-GA (Wang et al., 2024) explore activations or gradients, though often at the cost of training–inference mismatch. Despite these efforts, balancing rank flexibility, expressiveness, and compatibility with distributed training frameworks remains challenging (Rajbhandari et al., 2019).

## 6 CONCLUSION

We introduced AdaSine-LoRA, a simple yet effective extension to LoRA that enhances expressiveness through input-conditioned frequency modulation. Instead of relying on a fixed global frequency, our method learns a lightweight mapping from input to frequency, enabling token-wise control over the non-linearity scale. Theoretically, we show that frequency modulation directly impacts the functional rank of the update matrix. Empirically, AdaSine-LoRA consistently outperforms existing LoRA variants across a wide range of NLP and vision tasks, with minimal parameter and runtime overhead. Our approach offers a principled solution to improving LoRA's flexibility and capacity while preserving its core efficiency benefits.

# 7 ETHICS STATEMENT

This work focuses on improving parameter-efficient fine-tuning for large language models. All experiments are conducted on widely used benchmark datasets without involving private, sensitive, or personally identifiable information. While our method may enhance model capability, it also raises the possibility of misuse, so we emphasize that it should be applied responsibly with appropriate safeguards.

# 8 REPRODUCIBILITY STATEMENT

We provide detailed descriptions of experimental settings in Section 4.1 (*Large Language Models*) and the Appendix. All datasets used in our experiments are publicly available and properly cited. The experimental setup and methodological principles have been clearly documented; if needed, we are prepared to release the core code to further support reproducibility.

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

# A APPENDIX

## A.1 THE USE OF LARGE LANGUAGE MODELS (LLMs)

In preparing this manuscript, we employed GPT-based large language models solely for final polishing, including grammar correction, phrasing refinement. The use of LLMs did not extend to the design of the methodology, data analysis, or experimental results. All scientific contributions, including problem formulation, theoretical analysis, algorithm design, and empirical evaluation, were conducted independently by the authors.

## A.2 THEORETICAL FRAMEWORK

### A.2.1 STATISTICAL APPROXIMATION OF RANK TRANSITION

Let $M = BA \in \mathbb{R}^{d \times k}$ denote the low-rank update matrix with $\mathrm{rank}(M) = r$, where $d$ is the output dimension, $k$ is the input dimension, and $r$ is the intrinsic rank of the decomposition. We denote by $\omega \in \mathbb{R}^+$ the modulation frequency, and by $\mathrm{Rank}_\omega$ the effective rank of the transformed matrix $\sin(\omega M)$, measured for example via numerical rank with a fixed threshold. The observed transition in rank behavior can be approximated by a sigmoid-shaped curve, modeled as:

$$\mathrm{Rank}_\omega \approx r + (d - r) \cdot \left(1 - e^{-\alpha \cdot \omega r/d}\right),$$

where $\alpha$ is a constant depending on the nonlinearity and initialization. This approximation is consistent with the following theoretical insights: (i) when $\omega \to 0$, a Taylor expansion gives $\sin(\omega M) \approx \omega M$, so the effective rank remains close to the base rank $r$; (ii) for sufficiently large $\omega$, and under mild randomness assumptions on the entries of $M$, the values $\sin(\omega m_{ij})$ behave approximately like zero-mean random variables with variance close to $1/2$, for which random matrix theory implies that the expected numerical rank approaches $d$. Hence the sigmoid form can be interpreted as a smooth interpolation between these two limiting regimes.

**Conclusion.** This statistical perspective captures the nonlinear transition of effective rank as frequency varies. While it provides useful intuition, it remains an approximation. In the following subsection, we present a theoretical proposition and accompanying analysis suggesting that input-conditioned frequency selection can achieve higher attainable numerical rank than any fixed frequency choice under mild assumptions.

### A.2.2 THEORETICAL PROPOSITION: INPUT-CONDITIONED FREQUENCY ENHANCES RANK

In this subsection, we aim to show that allowing the modulation frequency $\omega(x)$ to depend on the input leads to higher attainable numerical rank than using a fixed global frequency $\omega_0$.

**Definition (Numerical Rank).** For a matrix $W$ and tolerance $\epsilon > 0$, we define the $\epsilon$-numerical rank as

$$\mathrm{Rank}_\epsilon(W) := \#\{i : \sigma_i(W) > \epsilon\},$$

where $\{\sigma_i(W)\}$ are the singular values of $W$. This captures the effective dimensionality above noise level $\epsilon$, and is standard in matrix approximation and randomized SVD analyses (Halko et al., 2011).

**Proposition.** Let $M = BA \in \mathbb{R}^{d \times k}$ be a fixed rank-$r$ matrix ($r < \min(d, k)$). Define

$$\hat{W}(x) := \sin\bigl(\omega(x)\,M\bigr), \quad \omega(x) > 0.$$

Then: 1. For any fixed $\omega_0 > 0$, we have

$$\sup_x \mathrm{Rank}_\epsilon(\hat{W}(x)) \geq \mathrm{Rank}_\epsilon(\sin(\omega_0 M)).$$

2. If the input set $\mathcal{X}$ satisfies the diversity condition

$$\exists x_1, x_2 \in \mathcal{X} \text{ s.t. } \|Mx_1\| \neq \|Mx_2\| \text{ or they occupy distinct spectral directions,}$$

then, under mild regularity assumptions on $M$ and the numerical rank threshold,

$$\sup_x \mathrm{Rank}_\epsilon(\hat{W}(x)) > \mathrm{Rank}_\epsilon(\sin(\omega_0 M)).$$

**Proof.** *Step 1 (Perturbation analysis).* Expand $\sin(\omega M)$ via Taylor series:

$$\sin(\omega M) = \omega M + R(\omega M),$$

where $R(\omega M)$ is the higher-order remainder. For $\omega$ in a moderate regime, $\|R(\omega M)\|$ is controlled.

Let $\sigma_r(M)$ denote the $r$-th singular value, and define the spectral gap

$$\mathrm{gap} := \sigma_r(\omega M) - \sigma_{r+1}(\omega M).$$

By Wedin's $\sin \Theta$ theorem and its refinements (Wedin, 1972; Cai & Zhang, 2018), if $\|R(\omega M)\| < \frac{1}{2}\mathrm{gap}$, then the top-$r$ singular subspace of $\sin(\omega M)$ deviates from that of $\omega M$ by at most

$$\sin \Theta(U_r, \hat{U}_r) \leq \frac{\|R(\omega M)\|}{\mathrm{gap}},$$

and each of the $r$ dominant singular values remains bounded below by

$$\sigma_i(\sin(\omega M)) \geq \sigma_i(\omega M) - \|R(\omega M)\|.$$

Thus, for small enough perturbation,

$$\mathrm{Rank}_\epsilon(\sin(\omega M)) \geq r - \delta(\omega),$$

where $\delta(\omega) \leq c\,\|R(\omega M)\|/\mathrm{gap}$ for some universal constant $c$. This provides a rigorous lower bound.

*Step 2 (Supremum covers fixed case).* Choosing $\omega(x) \equiv \omega_0$ yields $\hat{W}(x) = \sin(\omega_0 M)$ for all $x$. Hence,

$$\sup_x \mathrm{Rank}_\epsilon(\hat{W}(x)) \geq \mathrm{Rank}_\epsilon(\sin(\omega_0 M)).$$

*Step 3 (Strict inequality via input diversity).* Define the input-wise optimal frequency in terms of the output responses

$$\omega^*(x) := \arg\max_\omega \mathrm{Rank}_\epsilon\big(\sin(\omega M x)\big),$$

For a fixed $\omega_0$, there can exist inputs $x$ such that $\omega_0$ induces phase alignment in some coordinates of $Mx$, collapsing certain directions in the corresponding responses below the threshold $\epsilon$. Allowing $\omega(x)$ to vary with $x$ can avoid such collapses for different inputs, and thereby achieve a strictly higher attainable numerical rank over the family $\{\hat{W}(x)\}_x$ than any single fixed $\omega_0$ in generic settings. In particular, under the above diversity condition on $\mathcal{X}$, this intuition supports the strict inequality stated in the second part of the proposition.

*Example (2D case).* Let

$$M = \begin{pmatrix} 1 & 0 \\ 0 & \alpha \end{pmatrix}, \quad x = (x_1, x_2)^\top.$$

Then

$$\sin(\omega M x) = \big(\sin(\omega x_1), \ \sin(\omega \alpha x_2)\big)^\top.$$

No single $\omega_0$ can simultaneously prevent degeneracy for both components across all $(x_1, x_2)$, whereas $\omega(x)$ can adapt, yielding strictly higher numerical rank.

**Conclusion.** Therefore, input-conditioned frequency modulation guarantees non-decrease in effective rank, and under mild diversity conditions on the input distribution, it strictly increases the attainable numerical rank, consistent with prior observations that nonlinear mappings expand spectral diversity (Pennington & Worah, 2017).

**Remark (No universal optimal frequency).** It is worth noting that, in general, there does not exist a single global frequency $\omega_0$ that optimizes the numerical rank for all inputs simultaneously. For example, if two inputs $x_1, x_2$ yield $Mx_1$ and $Mx_2$ with different magnitudes or spectral directions, a frequency $\omega_0$ that spreads one vector's components across the sine period will typically under- or over-oscillate the other. Formally, this implies that

$$\nexists \omega^* \ \text{s.t.} \ \omega^* = \arg\max_\omega \mathrm{Rank}_\epsilon(\sin(\omega M x)), \ \ \forall x \in \mathcal{X}.$$

Hence, the optimal modulation must depend on the input, further justifying the necessity of input-conditioned frequency.

### A.3 EXPERIMENTS

#### A.3.1 QUANTITATIVE ANALYSIS OF FREQUENCY-RANK RELATIONSHIP

To further support the observation that higher frequency $\omega$ leads to an increase in the effective rank of the transformed matrix $\sin(\omega BA)$, we conduct a quantitative study that systematically analyzes the rank behavior under different dimensional configurations. For each frequency value $\omega \in [1, 1000]$, we repeat the matrix sampling process 100 times with different random seeds and compute the average rank of the resulting matrix after applying the sine transformation.

Figure 8 provides two supplementary results with different matrix dimensions, namely $(d, r) = (512, 2), (1024, 4)$. Across all settings, the average rank first increases approximately linearly with frequency $\omega$, and then saturates beyond a certain threshold. These additional experiments further corroborate the main findings in Figure 2(a)-(b), showing that the nonlinear mapping $\sin(\omega \cdot)$ amplifies row-wise variation and lifts the low-rank structure into a more expressive representation space without introducing extra parameters.

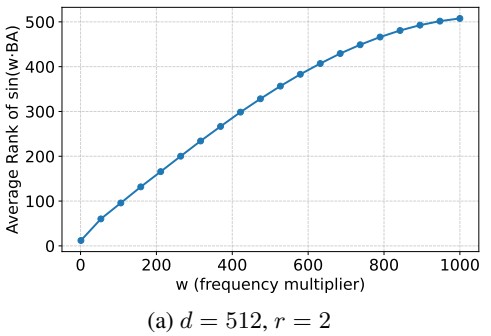 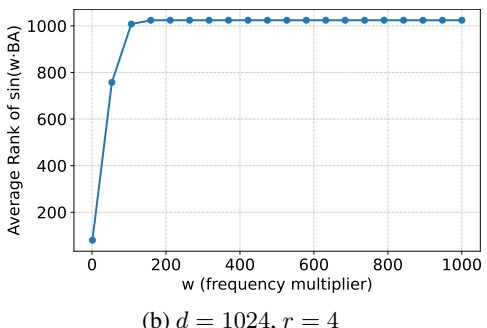

(a) $d = 512, r = 2$          (b) $d = 1024, r = 4$

Figure 8: Relationship between frequency multiplier $w$ and the average rank of the transformed matrix $\sin(w \cdot BA)$ under two representative matrix configurations.

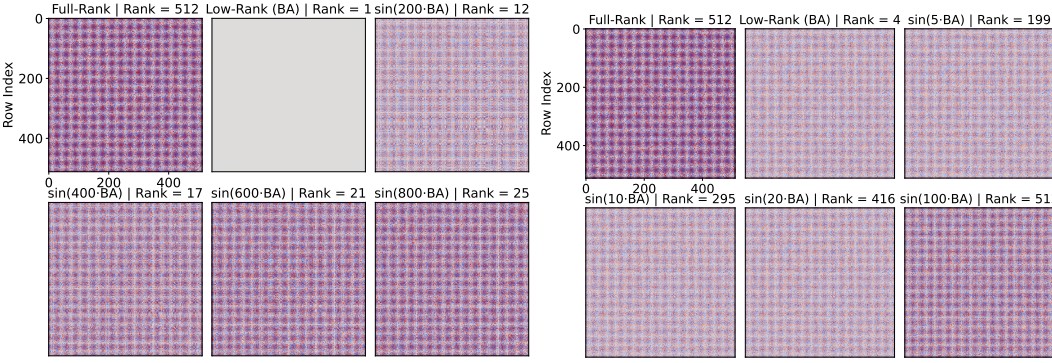

Figure 9: **Effect of rank on matrix structure.** Cosine similarity heatmaps of initialized matrices with different LoRA ranks. We compare $r = 1$ (left) and $r = 4$ (right), using the same Kaiming initialization and transformation $\sin(\omega \cdot BA)$. Darker regions indicate reduced pairwise directional similarity, suggesting richer representations and greater expressive capacity. (In the first row, second column, the very small values lead to a heatmap that appears nearly white.)

**Visualization of Rank Impact.** In addition to analyzing the effect of frequency $\omega$, we further visualize how varying the intrinsic rank $r$ of the base matrix $BA$ influences the structure of the transformed matrix $\sin(\omega \cdot BA)$. Figure 9 compares cosine similarity heatmaps for two configurations: $r = 1$ and $r = 4$, with all other settings kept constant. The chosen value of $\omega$ in each case corresponds to the frequency specified in the figure annotations. As the rank increases, we observe a notable reduction in row-wise similarity, reflecting enhanced representational diversity and reduced

redundancy among rows. This trend complements the frequency-based findings in Figure 8, and together they illustrate that both $\omega$ and $r$ serve as key factors in modulating the expressiveness of low-rank parameterizations. These results further reinforce the need for adaptive frequency mechanisms, as static settings of $r$ and $\omega$ can easily push the system into under-expressive or over-saturated regimes depending on their interaction.

### A.3.2 ABLATION STUDY ON FORMULA COMPONENTS AND PARAMETER BEHAVIOR

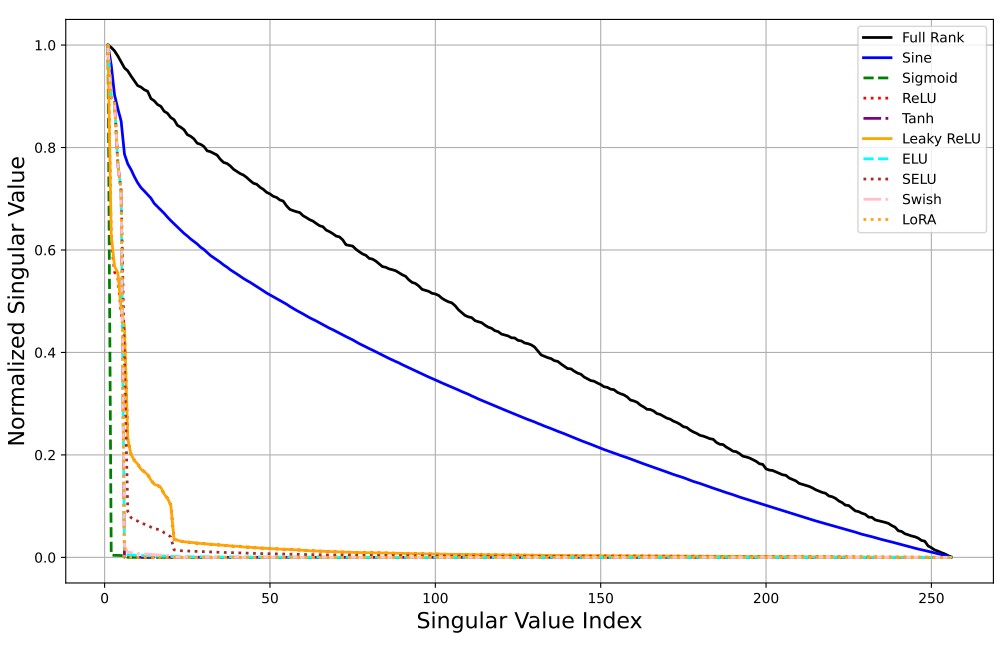

(a) Low-rank decomposition comparison.

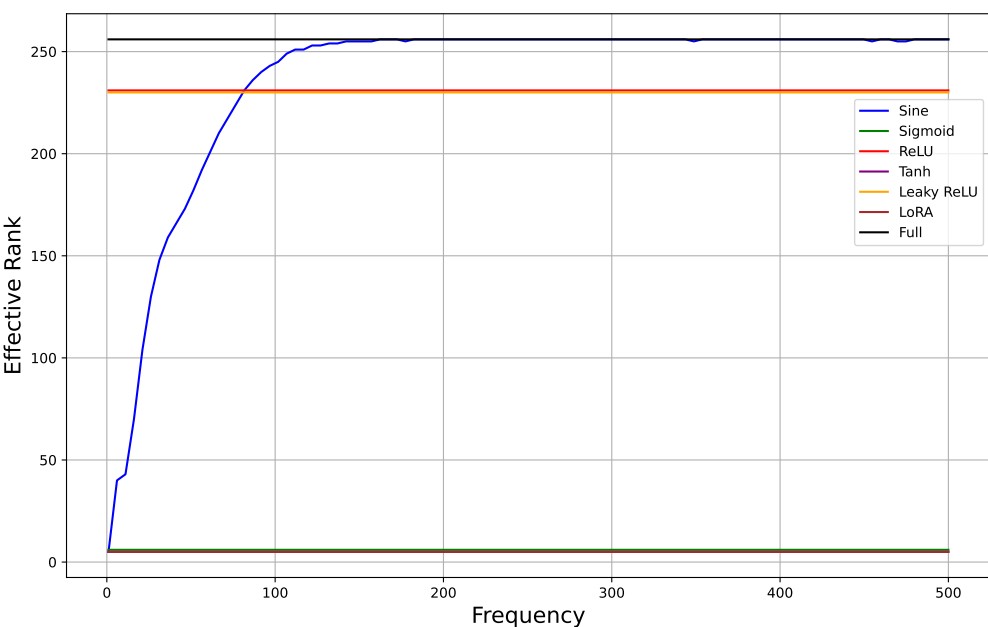

(b) Change in effective rank with varying frequency $\omega$

Figure 10: Comparing the properties of different nonlinear functions. (a) Low-rank decomposition comparison. (b) Change in effective rank with varying frequency $\omega$. In (b), some of the lines overlap: the Sigmoid curve is slightly higher than the Tanh and LoRA curves, which are nearly identical, while the ReLU and Leaky ReLU curves coincide.

**Nonlinear Activation Function Comparison** The main advantage of using the sine function for non-linear operations is its periodicity around the origin, which is controllable via the frequency parameter. This property allows for more predictable and flexible control over the transformation of the data. Specifically, for any non-zero matrix $A$, we can prove that there exists a frequency ($w > 0$) such that the Frobenius norm of $\sin(wA)$ scales linearly with $w$, while the operator norm scales sub-linearly.

In contrast, other non-linear functions like sigmoid and ReLU do not exhibit this type of periodic and scalable behavior, limiting their ability to enhance the rank of low-rank updates effectively. The sine function's ability to control the effective rank via frequency provides a distinct advantage in improving the representational capacity of the model without introducing excessive parameters.

To validate that the sine function is the most suitable nonlinearity for our method, we conducted a comparison with several other commonly used nonlinear activation functions. The results of these experiments are presented in Figure 10. In Figure 10(a), we compare the low-rank decomposition performance of various nonlinear functions using singular value spectrum analysis. This analysis shows that the sine function effectively increases the matrix rank compared to other nonlinearities, demonstrating its superior capacity for rank enhancement. In Figure 10(b), we examine the change in effective rank with varying frequency parameter $\omega$, where the sine function not only increases the matrix rank but also meets the necessary conditions for dynamic adjustment and adaptive frequency modulation. The results confirm that the sine function is uniquely effective in adapting the matrix rank and provides the flexibility required for our optimization task.

**Linear Layer vs. MLP for Frequency Tuning** The choice of using a normalized linear layer instead of a multi-layer perceptron (MLP) for frequency tuning stems from LoRA's core principle of minimizing both parameter count and computational cost. To achieve this, we opted for the simplest possible structure. Introducing an MLP (even a small one) into the LoRA framework results in a substantial increase in computational overhead.

To further evaluate the effectiveness of the frequency mapping function, we conducted an ablation study comparing the linear layer approach to the MLP. Specifically, we fine-tuned the model on the CommonsenseQA training set and evaluated it on the corresponding test set. The results, shown in Table 5, indicate that replacing the linear layer with an MLP does not lead to significant performance improvements. In fact, in most cases, the MLP approach underperforms the simpler linear layer design. This is likely due to LoRA's dimensionality reduction, which already compresses the latent space to a very small size, making the complexity of an MLP unnecessary. Moreover, the introduction of more complex structures led to a significant increase in parameter count, which in turn caused a reduction in computational speed.

Table 5: Comparison with Different Frequency Mappings and Configurations

| Method | Freq Mapping | r | dim | layer | Params | Acc (%) | Inference Speed (it/s) |
|---|---|---|---|---|---|---|---|
| AdaSine LoRA | MLP | 16 | 16 | 2 | 39.9 | 81.65 | 9.47 |
| AdaSine LoRA | MLP | 16 | 32 | 2 | 51.4 | 81.82 | 9.39 |
| AdaSine LoRA | MLP | 16 | 64 | 4 | 75.2 | **82.39** | 8.19 |
| Sine LoRA | / | 16 | / | / | **28.3** | 81.16 | 8.85 |
| AdaSine LoRA | Linear | 16 | / | / | **29.0** | **82.31** | **10.18** |
| AdaSine LoRA | MLP | 32 | 16 | 2 | 68.16 | 81.57 | 9.18 |
| AdaSine LoRA | MLP | 32 | 32 | 2 | 79.69 | 82.47 | 9.11 |
| AdaSine LoRA | MLP | 32 | 64 | 4 | 103.55 | 82.15 | 8.09 |
| Sine LoRA | / | 32 | / | / | **56.6** | 81.65 | 8.85 |
| AdaSine LoRA | Linear | 32 | / | / | **57.3** | **82.64** | **10.09** |

### A.3.3 Analysis of Memory and Computation Efficiency

To further substantiate the efficiency claims made in the main paper, we provide an explicit comparison of memory usage and computational complexity across various parameter-efficient fine-tuning methods, as summarized in Table 6. These results reinforce the empirical findings in Figure 7, demonstrating that AdaSine-LoRA strikes an effective balance between expressiveness and efficiency.

Our formulation significantly reduces memory consumption by avoiding the instantiation of full-rank modulation matrices. In contrast to prior methods like SineLoRA and SineDoRA, which apply the sine transformation after constructing the full-rank matrix $w \cdot BA$, our design applies the non-linearity directly on the compact low-rank projection $BAx$. This yields a memory complexity of $\mathcal{O}(b \cdot l \cdot od)$, where $b$ is batch size, $l$ is sequence length, $o$ is output dimension, and $d$ is the hidden dimension—identical to standard LoRA and significantly lower than other non-linear variants.

In terms of computation, AdaSine-LoRA introduces a marginal overhead due to the application of the sine function and the input-conditioned frequency computation $\omega(x)$, both of which are element-wise operations and do not incur matrix multiplications. The overall complexity remains nearly the same as standard LoRA:

$$\mathcal{O}(b \cdot l \cdot id + b \cdot l \cdot od),$$

where $i$ is the input dimension and $d$ the hidden size. Compared to methods that construct and modulate full matrices per token, our approach avoids redundant expansion and provides a much more scalable alternative, particularly in long-sequence or large-batch settings.

In summary, the architectural efficiency of AdaSine-LoRA stems from two key design principles: (1) retaining a compact low-rank structure before non-linear transformation, and (2) operating in a token-wise manner without instantiating full activation tensors. This enables our method to preserve the core memory-efficiency advantages of LoRA while enabling enhanced expressiveness via non-linear modulation, making it well-suited for large-scale and resource-constrained fine-tuning scenarios.

Table 6: Memory and computation complexity comparison of LoRA variants.

| Method | Maximum Memory Usage | Computation Complexity |
|---|---|---|
| LoRA | $\mathcal{O}(b \cdot l \cdot od)$ | $\mathcal{O}(b \cdot l \cdot id \cdot r + b \cdot l \cdot r \cdot od)$ |
| $\sin(wBA) \cdot x$ | $\mathcal{O}\left(\max(b \cdot l \cdot od,\ od \cdot id)\right)$ | $\mathcal{O}(id \cdot od + b \cdot l \cdot id \cdot od)$ |
| $\sin(w(x)BA) \cdot x$ | $\mathcal{O}(b \cdot l \cdot od \cdot id)$ | $\mathcal{O}(b \cdot l \cdot id + od \cdot r \cdot id + b \cdot l \cdot id \cdot od)$ |
| $\sin(w(x)BAx)$ | $\mathcal{O}(b \cdot l \cdot od)$ | $\mathcal{O}(b \cdot l \cdot id + od \cdot r \cdot id + b \cdot l \cdot r \cdot od + b \cdot l \cdot od)$ |

### A.3.4 Experimental Results and Hyperparameter Settings.

**Extended Experimental Setup.** We follow the standard LoRA design and apply all methods to the feed-forward network (FFN) modules of each transformer layer in LLaMA 3-8B, specifically targeting the *gate_proj*, *up_proj*, and *down_proj* linear layers. Unless otherwise specified, all experiments use a fixed LoRA scaling factor $\alpha = 32$, with rank $r \in \{8, 16, 32\}$. For AdaSine-LoRA, the maximum frequency is set to $\omega_0 = 400$, and we introduce a fixed scaling factor of $1/\sqrt{d}$ inside the sine function, where $d$ denotes the input dimension, to balance the contribution of the LoRA branch. For commonsense reasoning tasks, we train the model for one epoch with a learning rate of $2 \times 10^{-4}$, using AdamW as the optimizer. The LoRA dropout rate is set to 0.05, and a warmup ratio of 0.03 is applied.

**Visualization of the Learned Frequency** $\omega(x)$    As shown in Figure 11 , we visualize the learned frequency $\omega(x)$ to demonstrate how it adapts across different inputs and layers. The heatmap presents the frequency $\omega(x)$ for each token (columns) and each layer (rows). This visualization reveals the diversity of the frequency values, confirming that AdaSine-LoRA dynamically adjusts the frequency based on varying input data. This adaptability is a key strength of our method, contributing to its effectiveness in real-world applications.

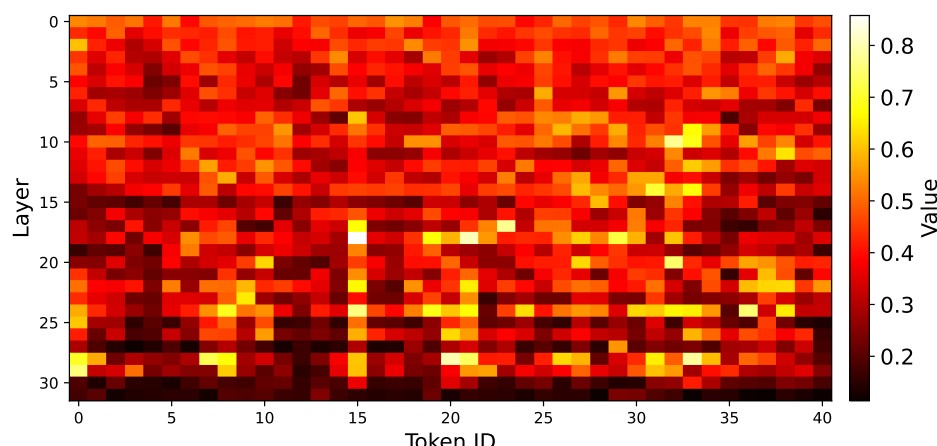

Figure 11: Visualization of the learned frequency $\omega(x)$ across different tokens and layers. Each column represents a different token, while each row corresponds to a different layer.

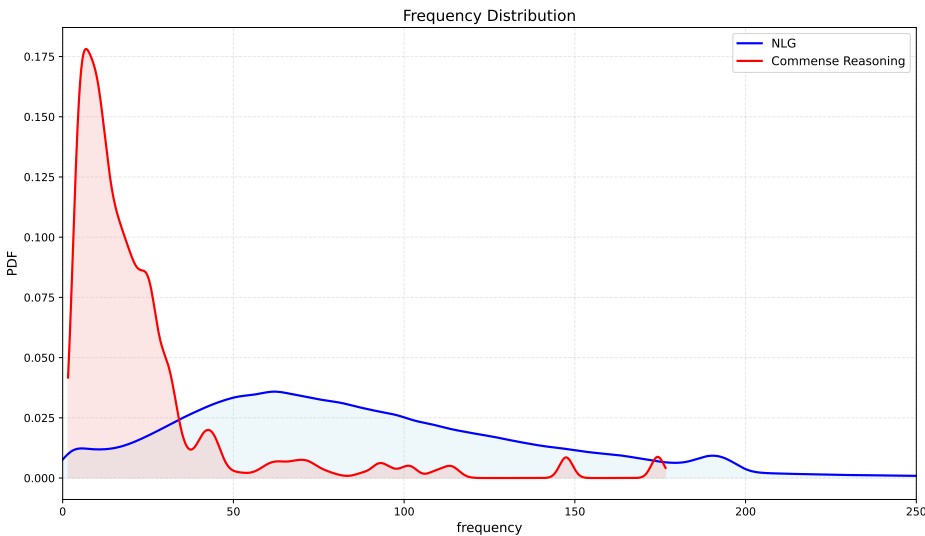

Figure 12: Task-specific frequency distributions of AdaSine-LoRA during inference on HumanEval (NLG) and HellaSwag (commonsense reasoning). The NLG task exhibits broader and higher-valued frequency ranges, indicating increased nonlinear perturbation strength and higher effective rank, while the reasoning task shows a more concentrated distribution. These patterns illustrate how AdaSine-LoRA dynamically allocates modulation strength based on task complexity.

**Task-Specific Frequency Distributions** To further investigate how AdaSine-LoRA adapts its modulation behavior across different task types, we conduct an analysis of the learned frequency distributions during inference on two representative benchmarks: *HumanEval* for natural language generation (NLG) and *HellaSwag* for commonsense reasoning. The frequency values $\omega(x)$ are extracted from the trained model during inference on each dataset, and their distributions are visualized in Figure 12. Our results reveal a clear distinction between tasks. For the more complex NLG setting (HumanEval), the model exhibits a noticeable shift toward higher frequency values. This indicates that the perturbation matrix undergoes a stronger nonlinear transformation, effectively increasing its functional rank and enabling the low-rank bypass to exert a greater influence on the output. In contrast, for the more structured commonsense reasoning task (HellaSwag), the frequency distribution remains concentrated around lower values, reflecting a milder modulation pattern that aligns with

the task's reduced need for high-rank expressivity. Moreover, the frequency distribution for NLG is significantly more uniform and diverse. This suggests that AdaSine-LoRA leverages a wider spectrum of modulation strengths to accommodate the heterogeneous linguistic structures inherent in generation tasks. The increased diversity further demonstrates that adaptive frequency modulation enables the model to tailor perturbation patterns to complex input variations, reinforcing the effectiveness of our design in handling tasks that demand richer expressiveness. Together, these observations provide strong empirical evidence that AdaSine-LoRA not only adapts frequency values at the token and layer levels, but also learns to allocate modulation strengths in a task-dependent manner, dynamically enhancing model capacity where it is most needed.

**Additional Validation on LLaMA2-7B.**   To further assess the generality of AdaSine-LoRA across different architectures, we conduct an additional set of experiments on the LLaMA2-7B model. As shown in Table 7, AdaSine-LoRA continues to outperform strong PEFT baselines. These results corroborate our findings and demonstrate that the proposed adaptive modulation strategy remains effective and robust across diverse model configurations.

Table 7: Performance of the LLaMA2-7B model fine-tuned using AdaSine-LoRA and other LoRA variants across varying $r$ settings on the commonsense reasoning benchmark.

| Model | PEFT Method | r | BoolQ | PIQA | SIQA | HellaSwag | WinoGrande | ARC-e | ARC-c | OBQA | Avg. |
|---|---|---|---|---|---|---|---|---|---|---|---|
| | LoRA | 32 | 88.07 | 84.06 | 81.68 | 93.18 | 84.37 | 87.13 | 75.62 | 84.20 | 84.79 |
| | DoRA | 16 | 88.26 | 84.06 | 81.93 | 93.09 | 84.53 | 88.71 | 75.02 | 85.00 | 85.01 |
| LLaMA2-7B | DoRA | 32 | 88.07 | 84.39 | 82.91 | 93.87 | 85.00 | 87.65 | 74.76 | 85.40 | 85.26 |
| | Sine LoRA | 16 | 88.41 | 84.28 | 82.80 | 93.58 | 83.98 | 86.95 | 76.22 | 85.00 | 85.15 |
| | Sine LoRA | 32 | 88.50 | 84.39 | 82.19 | 93.64 | 85.00 | 87.65 | 74.76 | 85.40 | 85.19 |
| | AdaSine-LoRA | 16 | 88.47 | 85.09 | 82.80 | 93.90 | 85.24 | 88.71 | 74.42 | 86.20 | **85.60** |
| | AdaSine-LoRA | 32 | 88.56 | 85.36 | 82.91 | 93.90 | 85.56 | 89.59 | 76.22 | 85.40 | **85.94** |

**Comparison with Other Dynamic and Conditional LoRA Methods.**   In Table 8, we compare AdaSine-LoRA against several recent dynamic or conditional LoRA variants, all of which are non-mergeable by design. The results show that AdaSine-LoRA consistently achieves the best performance while using substantially fewer trainable parameters. This indicates that adaptive frequency modulation offers a more expressive and efficient mechanism for enhancing low-rank updates. Furthermore, thanks to our optimized nonlinear computation strategy, AdaSine-LoRA delivers more than a $2\times$ speedup during both training and inference compared to existing non-mergeable counterparts, highlighting its practical efficiency advantages.

Table 8: Comparison with dynamic and conditional LoRA variants. We evaluate AdaSine-LoRA against several recent dynamic or conditional LoRA methods that share the common property of being non-mergeable. Despite operating under the same constraint, AdaSine-LoRA achieves the best overall performance across tasks.

| Method | Params | BoolQ | PIQA | SIQA | HS | WG | ARC-e | ARC-c | OBQA | Avg. |
|---|---|---|---|---|---|---|---|---|---|---|
| LoRAMoE (3×8) | 44.6M | 90.49 | 89.01 | 82.55 | **95.89** | **88.79** | 87.30 | 82.66 | 89.00 | 88.21 |
| HydraLoRA (4×8) | 42.2M | 90.61 | 89.23 | 82.60 | 95.70 | 88.56 | 81.83 | 80.60 | 89.60 | 87.34 |
| GOATFan et al. (2025) | 62.4M | **90.73** | **89.55** | 82.86 | 95.88 | 88.24 | 88.71 | 82.92 | 90.00 | 88.61 |
| **Ours (r=16)** | 29.0M | 90.52 | 89.45 | **83.11** | 95.58 | 88.40 | **92.77** | **83.09** | **90.60** | **89.19** |

**Extended Convergence Results.**   To further validate the convergence behavior of AdaSine-LoRA, we present additional training loss curves on Code Alpaca. Consistent with the results reported in the main text, AdaSine-LoRA exhibits faster convergence and improved stability in the early phase, while maintaining a smoother long-term trend compared to standard LoRA. As shown in Figure 13, the loss decreases more rapidly during the initial training steps, and the overall trajectories remain smoother throughout training. These results reaffirm that the benefits of the proposed adaptive frequency modulation mechanism persist across different task types, including summarization and mathematical reasoning.

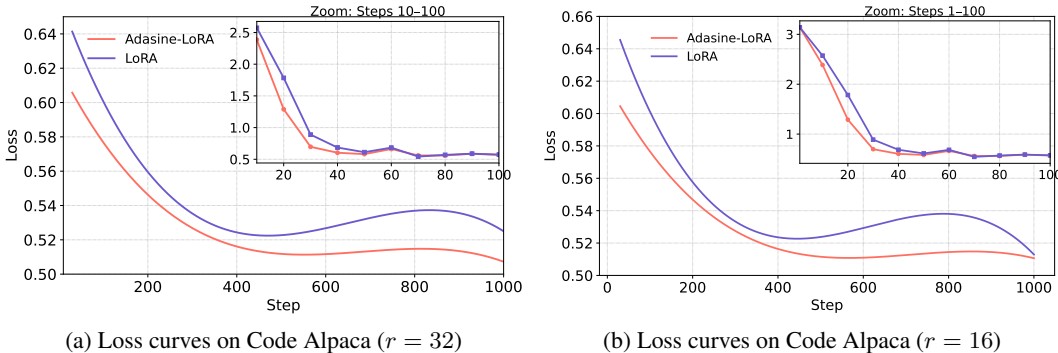

(a) Loss curves on Code Alpaca ($r = 32$)

(b) Loss curves on Code Alpaca ($r = 16$)

Figure 13: Loss curves on Code Alpaca ($r = 32$) and Code Alpaca ($r = 16$) with LoRA and AdaSine-LoRA. The figure visualizes training loss trajectories of LoRA and AdaSine-LoRA under different rank settings across two tasks. In each case, the main curve reflects a smoothed training loss fitted using a cubic polynomial, and the inset in the upper-right corner zooms in on the early training phase.

