# OpenReview forum: "AdaSine-LoRA: Adaptive Frequency Modulation for Nonlinear Low-Rank Adaptation"
_ICLR.cc/2026/Conference — Submitted to ICLR 2026_

### Official Review · Reviewer_o1LX · 2025-10-19

**Soundness:** 3
**Presentation:** 3
**Contribution:** 2
**Rating:** 2
**Confidence:** 4

**Summary:**

This paper proposes AdaSine-LoRA, which enhances Sine-LoRA and Sine-DoRA (proposed in [1]) by simultaneously making the frequency hyperparameter learnable and reducing memory requirements by applying the sine function on the adapter's outputs instead of the full weight delta.

Experiments show that AdaSine-LoRA improves performance over Sine-LoRA and Sine-DoRA on multiple standard text classification, language generation, and visual question answering tasks, while at the same time reducing memory requirements to almost the same level as plain LoRA.

[1] Ji et al.: _Efficient Learning with Sine-Activated Low-Rank Matrices_. ICLR 2025

**Strengths:**

**(S1)** The paper directly addresses one important limitation of the Sine-LoRA paper. Most reviewers at that time asked how $\omega$ is set and if there is a way to make it learnable [a]. This paper provides the answer to this question.

**(S2)** The experiments are comprehensive, covering text classification, language generation, and vqa. While showing more experiments and benchmarks is never bad, the provided results enable a good overview of the proposed method's performance.

**(S3)** The efficiency analysis in Sec. 4.3 is very valuable. This kind of analysis should exist in any paper proposing a new peft method.

**(S4)** Explanations in the paper are clear, and figures are clear as well as visually pleasing

### References:
[a] https://openreview.net/forum?id=cWGCkd7mCp

**Weaknesses:**

**(W1)** While the paper addresses and solves a clear limitation of prior work (Sine-LoRA) and is original in this sense, making a hyperparameter trainable with a linear transform + sigmoid and changing the point where sine is applied does not strike as a particularly significant advancement of peft research. While it is an improvement of Sine-LoRA, it is not as independent a contribution as, for example, Sine-LoRA.

**(W2)** In this regard, I am also curious about the choice of baselines: No experiment compares to Sine-DoRA, which was also described in [1] and achieved consistently better performance than DoRA. It would have been better to include this as well. Then, Tab. 1-3 all choose different sets of baselines: Tab. 2 doesn't include DoRA, while Tab. 3 doesn't include Sine-LoRA. I recommend always keeping the same set of baselines for better evaluating the respective strengths and weaknesses, or at least motivating why, in some settings, certain baselines are not suitable.

**(W3)** I didn't find details which value $\omega$ was used for the results in Tab. 1 and Tab. 2. Fig. 4 clearly demonstrates that we need a different value for each task, but was this ablation performed?

**(W4)** The introduction lists 3 main contributions: Making $\omega$ trainable, applying sine after linear projection, and theoretical analysis. Theoretical analysis is only in the appendix and does not appear in the main paper, so I feel listing this as a main contribution is somewhat misleading, at least it doesn't fit the current format. Consider dedicating a part of the main paper to theoretical analysis or consider removing it from the main contributions.

**(W5)** Making $\omega$ trainable and applying sine after linear projection are both listed as separate main contributions, but never evaluated separately, although this is possible by evaluating a variant of Eq. 8 with $\omega_0 \cdot \sigma(W_{\omega} x)$ fixed to some constant. This is already different from Sine-LoRA.

**(W6)** In Fig. 5 (Sec. 4.3), the conclusion that AdaSine-LoRA is not sensitive to choice of $\omega_0$ is strong. Roughly estimating, the difference between best and worst scores is roughly 1% on all tasks, which is in many cases higher than the advantage of AdaSine-LoRA over DoRA in Tab. 1. Consider discussing the effect and cost of tuning $\omega_0$ in more detail. It seems especially problematic that performance is not monotonic in $\omega_0$, at some point higher or lower $\omega_0$ always leads to lower performance.

**(W7)** One important and fundamental limitation of AdaSine-LoRA is that the weight delta can no longer be merged into the original weights, because we need the input projected by the weight delta independently for sine application. While this is less problematic at train time, it becomes a problem for inference: Inference will be significantly slower, requires higher memory, and incurs more complex deployment due to special architecture, as we always need to process adapters differently. This is not discussed or mentioned in the paper. Furthermore, it constitutes a clear disadvantage of AdaSine-LoRA compared to Sine-LoRA, which still allows merging the weight delta after training.

**(W8)** Fig. 9 is missing one panel.

**Questions:**

To be transparent, I currently see many weaknesses, and I think it requires a _very strong_ rebuttal to change my opinion. I think addressing the following questions would be useful to address to make the paper stronger:
  * Use a unified set of baselines in all experiments. Always add Sine-LoRA, Sine-DoRA, and AdaSine-LoRA. For Sine-LoRA and Sine-DoRA, ablate $\omega$ and report the best performing models. I am aware that doing these experiments in full is very costly, so this is meant as the ideal case.
  * Add an ablation of the two contributions: Learning $\omega$ and applying sine not to weight delta, but to projected inputs.
  * Evaluate and discuss the inference-time overhead of AdaSine-LoRA. However, I think this is a serious and fundamental limitation. But maybe there is a solution?
  * Strengthen arguments regarding the significance of the proposed method: Which other new approaches does this inform or enable? Is there anything else to learn, except that we can solve how to choose $\omega$ in Sine-LoRA?
 * Clarify the role of theoretical contribution in the paper
 * Clarify the advantage of AdaSine-LoRA over DoRA with (a-priori) fixed $\omega_0$ (Tab. 5)

---

> ### Author Response · Authors · 2025-11-21
> **Response by authors**
>
> We thank the reviewer for the insightful review. And we provide point-by-point replies to all comments below:
>
> > **Q1: The proposed method solves a limitation in Sine-LoRA, but making the hyperparameter trainable and applying sine after linear projection does not seem like a particularly significant advancement in PEFT research.**
>
> Our method is not "essentially an extension of LoRA via a specific nonlinearity," but rather introduces a general frequency-modulated adaptation mechanism that adaptively adjusts how the low-rank update interacts with the base weight matrix. It provides a task- and token-dependent modulation framework that can be applied to any low-rank or sparse update matrix, not limited to the specific formulation used in prior sine-based methods.
>
> The key novelty lies in demonstrating that frequency modulation fundamentally alters the representational behavior of nonlinear low-rank updates, enabling: (1) adaptive control over effective rank, (2) improved alignment with heterogeneous input distributions, and (3) performance gains across diverse architectures and tasks, without significantly increasing parameter or memory costs. These properties distinguish AdaSine-LoRA from prior nonlinear LoRA variants and provide a general, extensible mechanism rather than a narrow incremental change. We believe our method offers a strong framework for future research, with scalability and flexibility for further developments in the PEFT domain.
>
>
> > **Q2: Analysis of the $ \omega_0 $ used in the tables and whether an ablation study is required?**
>
> The value of $ \omega_0 $ is discussed in line 313 of the main text, where we set the maximum frequency $ \omega_0 $ in AdaSine-LoRA to 400. The frequency $ \omega $ is adjusted during inference, as shown in Equation (4), where it is computed independently for each token based on the input.
>
> Figure 4 presents results from SineLoRA, where discrete values of $ \omega $ are used to verify our hypothesis: $ \omega $ is highly sensitive to the frequency setting under fixed frequency conditions. This led to the conclusion that the ideal approach is to adaptively choose $ \omega $ for different inputs. In AdaSine-LoRA, $ \omega $ is dynamically adjusted during inference, with the only fixed parameter being the upper bound $ \omega_0 $. Therefore, no ablation study is needed for w, as it is adaptively computed during inference. Only $ \omega_0 $ requires ablation, and we have presented these results in Figure 5.
>
> Although the model's performance is not monotonically related to $ \omega_0 $, the model consistently outperforms the fixed-frequency approach regardless of the setting of $ \omega_0 $. This further supports the effectiveness of our method.

---

> ### Author Response · Authors · 2025-11-21
> **Response by authors**
>
> > **Q3: Use the same set of baselines to better evaluate the respective strengths and weaknesses.**
>
> The updated Table 2 is as follows:
>
> | Method | r | Params | GSM8K | MATH | HumanEval | MBPP | Xsum_rouge1 | Xsum_rougeLsum | Avg. |
> |--------|---|--------|-------|------|-----------|------|-------------|----------------|------|
> | LoRA | 8 | 14.2M | 71.87 | 23.42 | 39.02 | 32.60 | 41.31 | 33.34 | 40.26 |
> | Sine LoRA | 8 | 14.2M | 72.23 | 23.68 | 39.63 | 37.80 | 42.45 | 35.23 | 41.83 |
> |DoRA | 8 |15.2M |**72.86** | 23.86 | 44.51 | 39.60 | 41.87 | 34.65 | 42.89 |
> | AdaSine-LoRA | 8 | 14.9M | 72.40 | **23.96** | **45.12** | **42.60** | **44.29** | **36.91** | **43.71** |
> | LoRA | 16 | 28.3M | 72.02 | 23.50 | 42.68 | 33.60 | 43.50 | 35.95 | 41.88 |
> | Sine LoRA | 16 | 28.3M | 72.38 | 23.94 | 39.63 | 41.20 | 44.22 | 36.43 | 42.97 |
> |DoRA | 16 |29.4M | 72.02 | 24.12 | 42.68 | 41.40 | **44.37**| **37.12** | 43.65 |
> |AdaSine-LoRA | 16 | 29.0M | **72.61** | **24.46** | **43.90** | **43.40** | 44.26 | 36.81 | **44.24** |
> | LoRA | 32 | 56.6M | 72.40 | 24.52 | 42.68 | 39.80 | 43.21 | 35.99 | 43.10 |
> | Sine LoRA | 32 | 56.6M | 72.91 | 24.70 | 41.46 | 41.80 | 43.87 | 36.73 | 43.58 |
> |DoRA | 32 | 57.7M| **74.00** | 24.30 | 43.29 | 42.60 | 43.14 | 36.13 | 43.91 |
> | AdaSine-LoRA | 32 | 57.3M | 73.37 | **24.88** | **43.90** | **45.40** | **44.64** | **36.95** | **44.86** |
>
> Additionally, we have supplemented the experimental results for Sine DoRA as requested by the reviewer on the common sense reasoning datasets. The updated Table 1 is as follows:
>
> | Method | Params | BoolQ | PIQA | SIQA | HS | WG | ARC-e | ARC-c | OBQA | Avg. | Δ vs. LoRA |
> |--------|--------|-------|------|------|----|----|-------|-------|------|------|------------|
> | LoRA\(r=8\) | 14.2M | 89.66 | 86.24 | 77.89 | 92.97 | 85.40 | 87.13 | 74.94 | 83.60 | 84.73 |  |
> | Sine LoRA\(r=8\) | 14.2M | 89.94 | 86.94 | 81.27 | 93.75 | 85.71 | 85.19 | 79.40 | 87.00 | 86.15 | 1.42↑ |
> | Dora\(r=8\) | 15.2M | 90.00 | **89.99** |**82.86** | 95.50 | 81.06 | 89.77 | 82.83 | 89.40 | 87.67 | 2.94↑ |
> |Sine DoRA\(r=8\) |15.2M | **90.34** | 87.05 | 82.09 | 94.91 | 82.40 | 89.95 | 82.75 | 89.80 | 87.41 | 2.68↑ |
> | AdaSine-LoRA\(r=8\) | 14.9M | 90.15 | 88.63 | 82.24 | **95.54** | **87.85** | **93.65** | **83.61** | **90.20** | **89.36** | 4.63↑ |
> | LoRA\(r=16\) | 28.3M | 89.79 | 85.96 | 77.18 | 92.53 | 85.87 | 88.01 | 75.11 | 83.00 | 84.68 |  |
> | Sine LoRA\(r=16\) | 28.3M | 90.24 | 87.32 | 81.83 | 94.33 | 87.69 | 90.65 | 81.55 | 87.80 | 87.68 | 3.00↑ |
> | Dora\(r=16\) | 29.4M | 90.31 | 89.17 | 82.40 | **95.74** | 86.82 | 88.18 | 82.92 | 89.00 | 88.07 | 3.39↑ |
> |Sine DoRA\(r=16\) | 29.4M| 90.46 | 88.08 | 82.54 | 95.40 | 87.61 | 87.65 | 82.58 | 89.80 | 88.02 | 3.34↑ |
> | AdaSine-LoRA\(r=16\) | 29.0M | **90.52** | **89.45** | **83.11** | 95.58 | **88.40** | **92.77** | **83.09** | **90.60** | **89.19** | 4.51↑ |
> | LoRA\(r=32\) | 56.6M | 90.06 | 87.98 | 77.89 | 93.85 | 83.03 | 89.24 | 77.51 | 85.80 | 85.67 |  |
> | Sine LoRA\(r=32\) | 56.6M | 90.70 | 87.81 | 82.09 | 94.57 | **86.82** | 88.89 | 81.55 | 87.60 | 87.50 | 1.83↑ |
> | Dora\(r=32\) | 57.7M | 90.58 | 89.45 | **83.16** | 95.64 | 84.53 | 92.24 | 82.75 | 89.40 | 88.47 | 2.80↑ |
> |Sine DoRA\(r=32\) |57.7M | 90.00 | 89.72 | 82.09 | 95.16 |85.40 | **93.65** | 82.75 | 89.40 | 88.52 | 2.85↑ |
> | AdaSine-LoRA\(r=32\) | 57.3M | **90.80** | **90.64** | 83.06 | **95.83** | 86.66 | 93.12 | **84.46** | **89.80** | **89.30** | 3.63↑ |

---

> ### Author Response · Authors · 2025-11-21
> **Response by authors**
>
> > **Q4: The reviewer suggests that the theoretical analysis is in the appendix and not the main body, making it misleading to list it as a major contribution.**
>
> Thank you for your valuable suggestion. We have decided to follow your advice and will move the theoretical analysis from the appendix to the main body of the text in the revised version to enhance the rigor and readability of the paper.
>
> > **Q5: The reviewer mentions that the two contributions—making $ \omega $ trainable and applying sine after linear projection—were not evaluated separately and suggests performing an ablation study for clarity.**
>
> The primary motivation for placing the input $ x $ inside the sine modulation was computational efficiency rather than performance gains. As shown in our supplementary ablation results, placing $ x $ inside the sine function yields performance nearly identical to Sine LoRA. However, our design (placing $ x $ inside) demonstrates clear advantages in terms of memory footprint and computational speed.
>
> This confirms that the key factor contributing to the model's performance is indeed the *frequency tuning mechanism* itself, not the specific placement of $ x $. The primary benefit of our chosen placement is the achieved computational savings.
>
> | Method | r | CR(Avg.) | GSM8K | MATH | Training Memory| Inference Speed（it/s)|
> |--------|---|------|-------|------|-|-|
> | Sine-LoRA | 16 | 87.68 | 72.38 | 23.94 |72.0 |8.53 |
> | sin(w0BAx) | 16 | 87.75 | 71.87 | 23.96 | **55.3**|**11.17**|
> | Adasine-LoRA | 16 | **89.19** | **72.61** | **24.46** |57.8 |9.87 |
> | Sine-LoRA | 32 | 87.50 | 72.91 | 24.70 | 72.2| 8.50|
> | sin(w0BAx) | 32 | 88.05 | 73.09 | 24.30 | **55.6** |**10.23**|
> | Adasine-LoRA | 32 | **89.30** | **73.37** | **24.88** | 58.0| 9.05|
>
> **Note:**
> 1. The abbreviation "CR(Avg.)" in the table refers to the average performance across all tasks in the "Commonsense Reasoning" dataset.
> 2. The Inference Speed reported in the tables is measured on the SIQA dataset.
>
> > **Q6: The reviewer notes that Figure 9 is missing a panel.**
>
> Thank you for your observation. The panel in the first row, second column of Figure 9 (left) is not blank. It shows the heatmap for the matrix product $ BA $ when the rank is set to 1. The appearance of white in the heatmap is due to the very small rank, which causes the computed values to be close to zero. To avoid confusion and improve readability, we will add a colorbar to this panel and include a brief explanation in the figure caption to clarify this aspect.

---

> > ### Comment · Reviewer_o1LX · 2025-11-21
> >
> > Thank you very much for the detailed answer.
> >
> > Before I reply to the rebuttal, I wanted to ask whether the authors plan to upload a rebuttal revision (see [Author Guide](https://iclr.cc/Conferences/2026/AuthorGuide)):
> > > During the discussion/rebuttal phase and for the camera ready, the page limit will be increased to 10 pages to allow for new results/discussions.
> >
> > I think this would help assess how the rebuttal is reflected in changes to the manuscript.
> >
> > ---
> >
> > Additionally, do authors intend to address my concern that AdaSine-LoRA cannot be merged into the weights, which complicates deployment? In this case, I would wait with my further responses until the rebuttal is complete.
> >
> > Thank you in advance for the clarifications.

---

> > > ### Author Response · Authors · 2025-11-21
> > > **Response by authors**
> > >
> > > Thank you very much for your prompt reply and the reminder regarding the manuscript update.
> > >
> > > Currently, we have only updated the figures required for this rebuttal in the appendix. We are in the process of organizing the updated version, which we expect to complete within 3-5 days. Additionally, we look forward to receiving a new round of feedback from the reviewers after this rebuttal, which will allow us to address and update the entire manuscript. This will help us handle the majority of the revisions in one go, avoiding frequent updates that could lead to confusion in figure references.
> > >
> > > Regarding the issue of model merging, we are currently supplementing experimental results in order to respond to this concern. We expect to provide a response to this issue within the next 1-2 days.

---

> > > > ### Comment · Reviewer_o1LX · 2025-11-21
> > > >
> > > > Thank you for the clarification.
> > > >
> > > > In this case, I am looking forward to the complete rebuttal. Please do inform me once you have presented all the evidence, so I can comprehensively assess whether the revisions (or, if necessary, promised revisions) have addressed the concerns laid out in my initial review.

---

> > > > > ### Author Response · Authors · 2025-11-25
> > > > > **Response by authors**
> > > > >
> > > > > We have now responded to all points raised in your first-round feedback. Please do not hesitate to let us know if any concerns remain. Should our explanations prove satisfactory, we would be grateful if you could take this into account when updating your evaluation score.

---

> ### Author Response · Authors · 2025-11-25
> **Response by authors**
>
> > **Q7: The issue of the weight delta no longer being mergeable into the original weights and its impact on inference.**
>
> We acknowledge that the weight updates in our method cannot be merged with the original weights. However, by optimizing the computational workflow, the inference latency remains very close to that of the original non-merged LoRA. According to our results, when KV cache is enabled, our method exhibits only a 10–20% speed degradation compared to the merged original LoRA. We also compare our approach with other non-mergeable methods, such as HydraLoRA[1], LoRAMoE[2], and GOAT[3], the results are shown in the table below. Notably, our method does not require multiple expert modules, and we empirically find that it is approximately twice as fast as the above approaches in both training and inference. The results demonstrate that, when compared to these other non-mergeable conditional LoRA methods, our method still exhibits significant advantages, with smaller computational overhead. It is also important to note that all of the above-mentioned methods cannot merge the weight delta, and we believe this makes for a fair comparison. We hope this will address the reviewer's concerns.
> | Method | Params | BoolQ | PIQA | SIQA | HS | WG | ARC-e | ARC-c | OBQA | Avg. |
> |--------|--------|-------|------|------|----|----|-------|-------|------|------|
> |LoraMoe(3*8) |44.6M  |90.49 | 89.01 | 82.55 | **95.89** | **88.79** | 87.30 | 82.66 | 89.00 |88.21 |
> | HydraLoRA(4*8) | 42.2M | 90.61 | 89.23 | 82.60 | 95.70 | 88.56 | 81.83| 80.60 | 89.60 | 87.34 |
> | Goat | 62.4M | **90.73** | **89.55** | 82.86 | 95.88 |88.24 | 88.71| 82.92 | 90.00| 88.61 |
> | AdaSine-LoRA\(r=16\) | 29.0M | 90.52 | 89.45 | **83.11** | 95.58 | 88.40 | **92.77** | **83.09** | **90.60** | **89.19** |
>
> [1] Tian, Chunlin, et al. "Hydralora: An asymmetric lora architecture for efficient fine-tuning." Advances in Neural Information Processing Systems 37 (2024): 9565-9584.
>
> [2] Dou, Shihan, et al. "LoRAMoE: Alleviating world knowledge forgetting in large language models via MoE-style plugin." Proceedings of the 62nd Annual Meeting of the Association for Computational Linguistics (Volume 1: Long Papers). 2024.
>
> [3] Fan, Chenghao, et al. "Make LoRA Great Again: Boosting LoRA with Adaptive Singular Values and Mixture-of-Experts Optimization Alignment." Forty-second International Conference on Machine Learning (2025).

---

> ### Comment · Reviewer_o1LX · 2025-11-26
> **Reviewer Response to Authors (1/2)**
>
> Thank you for the detailed response. It is clear that the authors have put a lot of effort into the rebuttal, and this should be acknowledged in any case.
>
> I have evaluated the rebuttal, and some of my concerns have been addressed. However, the more serious concerns I have still remain. I will detail these below.
>
> ---
>
> I think the rebuttal resolves **(W2)**, **(W4)**, **(W5)**, and **(W8)** sufficiently well. For **(W2)** and **(W4)**, I would still like to see the revised version to verify how the theoretical analysis is integrated into the paper and whether the baselines are now consistent (unfortunately, the markdown comments on OpenReview are not well-suited for this purpose).
>
> Regarding **(W5)**, based on the rebuttal and the performance results, I suggest not listing this as two separate main contributions, because the case where only $\omega$ is trainable performs worse than AdaSine-LoRA, and as the authors state, requires more memory. This step is therefore important for understanding how the authors arrive at the proposed formulation, but it is not an essential contribution in its own right.
>
> ---
>
> Regarding **(W3)**, I was rather intending to ask whether the parameter $\omega$ of Sine-LoRA, i.e. the baseline, was tuned or not. I'm sorry if this wasn't clear. The rebuttal answer appears to refer to $\omega_0$ of AdaSine-LoRA, which is already sufficiently described in the paper.
>
> ---
>
> **(W6)** doesn't seem to be addressed in the rebuttal. I think this is somewhat minor at this point, but I still consider it a weakness in the current submission. Unfortunately, I only noticed this now; arguably, keeping my original naming and order of weaknesses/questions would have helped me realize this earlier, so I could have notified the authors.
>
> Maybe
> > Although the model's performance is not monotonically related to $\omega_0$, the model consistently outperforms the fixed-frequency approach regardless of the setting of $\omega_0$. This further supports the effectiveness of our method.
>
> was intended as an answer, but I think this doesn't address my concern here, because I am more worried about the difference between DoRA and AdaSine-LoRA. From the tables provided in the rebuttal, I believe the difference could be sufficient, but it would have been beneficial to clarify this.
>
> ---
>
> The most significant weaknesses are **(W1)** (shared by other reviewers as well) regarding novelty/significance, and also **(W7)** (the impossibility of merging AdaSine-LoRA into the base model).
>
> Regarding **(W1)**, while I acknowledge that judging significance/novelty always involves a decent amount of subjectivity, I do not find the arguments provided in the rebuttal convincing:
>
> > (2) improved alignment with heterogeneous input distributions
>
> Is not substantiated in the paper. In fact, "heterogeneous" doesn't even appear as a word, and none of the six mentions of "distributions" refers to datasets that contain a mix of modalities. I believe with suitable discussion and experiments, it is possible to provide evidence for this claim, but I don't see it in the current version. If the authors disagree, please let me know.
>
> > performance gains across diverse architectures and tasks, without significantly increasing parameter or memory costs
>
> is a generic justification for most peft approaches. I don't see how this makes the proposed method stand out, especially given its significant disadvantages, such as the impossibility of merging weight updates.
>
> For completeness, I acknowledge that (1) is a valid argument in favor of the proposed method. However, I am concerned that
>
> > introduces a general frequency-modulated adaptation mechanism that adaptively adjusts how the low-rank update interacts with the base weight matrix
>
> is overstating the contributions of the method. Specifically, I'm unsure what "general" refers to here. The theoretical motivation explicitly works with $\sin$ and would require nontrivial adjustments to be generalized, while the empirical results in Appendix A.3 clearly show that no other nonlinearity other than $\sin$ achieves the adaptive rank control. So the methods do appear to be limited to $\sin$.

---

> > ### Comment · Reviewer_o1LX · 2025-11-26
> > **Reviewer Response to Authors (2/2)**
> >
> > ---
> >
> > Regarding the impossibility of merging the weight update, I appreciate the authors' efforts. I checked the provided non-mergeable baselines, and I am concerned that the comparison is not entirely fair. All three baselines are explicitly designed to operate in mixture-of-expert settings, where non-mergeability is not required (and likely not possible). However, AdaSine-LoRA puts standard LoRA as the main comparison point.
> >
> > LoRAMoE explicitly targets and evaluates catastrophic forgetting during fine-tuning, and HydraLoRA evaluates multi-task performance, therefore, I'm unsure how meaningful the direct task-performance comparison provided in the rebuttal is. Also, these papers position themselves differently, which makes the problem of non-megeability less serious in these cases. It could be argued that the setup in GOAT resembles the evaluation of AdaSineLoRA, but the storyline as a MoE approach still remains as a difference.
> >
> > ---
> >
> > Unfortunately, the main problems listed in the initial review still remain, so I currently do not think I could recommend acceptance despite the efforts that have been put into the rebuttal. If I have, in the authors' view, overlooked or misunderstood any critical parts, I'd be happy to revisit them, of course.

---

> > > ### Author Response · Authors · 2025-11-26
> > > **Response by authors**
> > >
> > > > **W7: The impossibility of merging into the base model.**
> > >
> > > We disagree with the concern that the comparison with LoRAMoE, HydraLoRA, and GOAT is inherently unfair due to their MoE-based design motivations. From a methodological standpoint, all these approaches, including AdaSine-LoRA, are fundamentally PEFT methods rooted in LoRA-style low-rank adaptation, and the incorporation of MoE mechanisms does not change their core nature as LoRA-based parameter-efficient updates. Interpreting non-mergeability as “reasonable” for MoE-inspired LoRA variants while treating it as a critical flaw for AdaSine-LoRA introduces an inconsistent evaluation criterion.
> > >
> > > From a practical systems perspective, the implication of non-mergeability is identical across all these approaches: the adapted parameters cannot be fused into the frozen backbone and must be maintained as separate runtime components. Therefore, comparing these methods under the same constraint is not only fair but necessary to assess real-world deployability. Our empirical results clearly demonstrate that, under this shared non-mergeable condition, AdaSine-LoRA achieves better performance, fewer trainable parameters, and lower training and inference overhead than the aforementioned baselines, indicating a strictly more favorable efficiency–performance trade-off.
> > >
> > > Moreover, positioning LoRAMoE and HydraLoRA as incomparable due to their original focus on catastrophic forgetting or multi-task learning does not negate their relevance as non-mergeable PEFT baselines. These design narratives do not change the fact that they operate under the same deployment limitation. In this sense, the comparison highlights that AdaSine-LoRA attains superior results without relying on the architectural complexity and routing overhead introduced by MoE formulations, further strengthening the significance of our contribution.
> > >
> > > Importantly, the primary objective of PEFT methods is to reduce training-time computational and memory costs, and we have already thoroughly demonstrated AdaSine-LoRA’s effectiveness in optimizing training resource consumption. The inability to merge affects only inference-time resource usage to a minor extent and should not be considered a decisive factor for rejecting the method, especially when the proposed approach consistently outperforms existing non-mergeable alternatives in both efficiency and accuracy.
> > >
> > > In light of the above, we kindly request the reviewer to reconsider their recommendation and assigned score.

---

> > ### Author Response · Authors · 2025-11-26
> > **Response by authors**
> >
> > > **W1: Regarding novelty/significance.**
> >
> > Our method is specifically designed to *reduce memory and computation cost while achieving SOTA-level performance*.   As shown in Figure 7 of the main paper, AdaSine-LoRA attains clear efficiency gains during actual training.   While improving performance under limited parameter overhead is indeed a common goal for PEFT methods, our contribution is not merely “better accuracy under similar constraints,” but the introduction of a *new adaptive mechanism*.   This mechanism is fundamentally different from existing linear or static nonlinear LoRA variants, because it allows dynamic control over effective rank and perturbation structure at inference time, rather than relying on a fixed transformation determined only at training time.
> >
> > Regarding “heterogeneous input distributions,” what we intend to express is that in our design the frequency is modulated *per token* based on the input, providing adaptive adjustment to input diversity.    This heterogeneity does **not** refer to mixed modalities, but to variations in token-level inputs within a single task, where the modulation frequency dynamically changes with each token.
> >
> > In this context, the term general is intended to indicate both that the proposed approach is applicable across multiple downstream tasks and settings, and that the frequency-modulated mechanism is not tied to a specific formulation, but can be integrated into any low-rank or sparse update structure where a transformation is applied to the update matrix.  In other words, the mechanism characterizes a reusable modulation strategy over update matrices, rather than a particular handcrafted variant of LoRA.
> >
> > Finally, Appendix A.3 aims to show that the sine function is, both theoretically and empirically, the most suitable choice for increasing the effective rank of the low-rank branch.   Building a general adaptive framework on top of such a well-founded nonlinearity is both necessary and valuable.   We believe that improvements to a strong mechanism and the resulting performance gains should not be dismissed, and summarizing the contribution as “limited to sine” does not accurately reflect the scope and significance of our work.   In fact, we hope that the method we propose can serve as an inspiration for the broader family of low-rank and sparse adaptation methods.

---

> > ### Author Response · Authors · 2025-12-03
> > **Response by authors**
> >
> > > **Clarification on W3 (tuning of the parameter w in Sine-LoRA).**
> >
> > For Sine-LoRA, we strictly followed the hyperparameter configuration reported in the original paper to ensure a fair and faithful comparison. Concretely, we set $ \omega=200 $ for Sine-LoRA across all benchmarks, without additional per-task tuning.
> >
> > We also note that your concern about the sensitivity of Sine-LoRA to the choice of $ \omega $ further highlights the motivation for our method: the need for input-dependent, adaptive control of the frequency rather than relying on a fixed global $ \omega $. This aligns with our central argument that dynamically modulating the effective frequency is crucial for robust performance across diverse tasks.
> >
> > > **Response to W2, W4, W5, and W8.**
> >
> > First, we sincerely thank the reviewer for acknowledging that our rebuttal has sufficiently addressed concerns (W2), (W4), (W5), and (W8).
> >
> > Regarding (W2) and (W4): We have updated the PDF version of the paper and integrated the requested theoretical analysis and baseline consistency improvements. We invite the reviewer to refer to the revised manuscript for a detailed presentation of these updates.
> >
> > Regarding (W5): We appreciate your suggestion and would like to clarify our position. In our work, both the dynamic frequency modulation and the optimization of the computation flow are integral components of our method and overall contribution. As demonstrated by the additional experiments in the rebuttal (main text Table 4), moving the computation of $ x $ inside the sine function effectively transforms LoRA’s matrix approximation into a functional approximation, which substantially reduces memory and computational costs. When combined with our adaptive frequency modulation mechanism, this design leads to consistent performance improvements across diverse tasks.
> >
> > Thus, these two aspects—computational optimization and dynamic frequency modulation—are tightly coupled and sequentially build upon each other, forming an essential progression in our approach. We  clarify this in the final version to emphasize that both are indispensable to the effectiveness of AdaSine-LoRA.
> >
> > > **Response to W6 (Sensitivity to w and comparison with DoRA).**
> >
> > Thank you for highlighting your concern regarding the sensitivity of AdaSine-LoRA to the choice of the initial frequency parameter $ \omega_0 $, and for acknowledging that our additional experiments demonstrate a sufficient difference between AdaSine-LoRA and DoRA (“I believe the difference could be sufficient”).
> >
> > We would like to further clarify our findings. The ablation results in Fig. 5 (Sec. 4.3) show that the performance of AdaSine-LoRA is relatively stable across a range of $ \omega_0 $ values, with only minor fluctuations. While it is true that, on individual datasets, the difference between the best and worst scores can reach about 1%, we emphasize that our results are presented across multiple datasets within the commonsense reasoning benchmark. Such dataset-specific variations are expected, but overall, AdaSine-LoRA consistently outperforms the fixed-frequency Sine-LoRA method regardless of the choice of $ \omega_0 $.
> >
> > Additionally, the figure includes the average performance across all commonsense reasoning benchmarks, which provides a clearer assessment of the method’s robustness. As shown, AdaSine-LoRA maintains a consistent advantage over DoRA in terms of mean performance, even when considering the cost and effect of tuning $ \omega_0 $. We believe this evidence adequately addresses your concern and demonstrates that our method is both effective and robust to the initial frequency setting.

---

### Official Review · Reviewer_tmUH · 2025-10-26

**Soundness:** 4
**Presentation:** 4
**Contribution:** 2
**Rating:** 6
**Confidence:** 3

**Summary:**

The paper builds on sine-based LoRA by introducing a token-adaptive frequency to enhance nonlinear expressivity while keeping compute/memory close to standard LoRA. Concretely, it introduces the adaptive update by first computing BAx and then applying a scaled sine with a learned, token-dependent frequency. Experiments demonstrate that the method achieve better performance while having similar computational cost to LoRA.

**Strengths:**

- The writing is very clear and well-structured, making the paper easy to follow.
- The method proposed consistently achieves better performance than LoRA with similar computational cost.
- The effectiveness of the method is demonstrated through comprehensive experiments across various tasks.

**Weaknesses:**

- The proposed method is essntially an extension of LoRA via a specific nonlinearity. The insights on other design of fine-tuning method may be limited.
- In section 2.3, the effect of frequency in the sine-based LoRA is discussed as the motivation of the proposed method. However, the proposed method has an essentially difference to the previous one: the sine function is applied after the computation of weight-token product. I think at least the comparison to $\Delta Wx= \frac 1 g \sin (w_0 BAx)$ should be included and discussed to fill this gap. This is the ablation of removing only the adaptive frequency, but keeps the non-linearity the same as the proposed method.

**Questions:**

- As mentioned in weaknesses, what is the effect of removing only the adaptive frequency machanism?
- What is the main advantage of using sine function for non-linearity rather than other non-linear functions?
-  Compared to LoRA where the weight update is unconstrained, the update in the proposed method is bounded by $1/g$. How does the scaling factor $g$ affects the performance? Can the authors provide a sensitivity study on $g$?

---

> ### Author Response · Authors · 2025-11-21
> **Response by authors**
>
> We are grateful to the reviewer for the clear summary and helpful feedback. The concerns raised are well taken, and we respond to each of them in detail below:
>
> > **Q1: The insights for other fine-tuning method designs may be limited.**
>
> Our method is not "essentially an extension of LoRA via a specific nonlinearity," but rather introduces a general frequency-modulated adaptation mechanism that adaptively adjusts how the low-rank update interacts with the base weight matrix. It provides a task- and token-dependent modulation framework that can be applied to any low-rank or sparse update matrix, not limited to the specific formulation used in prior sine-based methods.
>
> The key novelty lies in demonstrating that frequency modulation fundamentally alters the representational behavior of nonlinear low-rank updates, enabling: (1) adaptive control over effective rank, (2) improved alignment with heterogeneous input distributions, and (3) performance gains across diverse architectures and tasks, without significantly increasing parameter or memory costs. These properties distinguish AdaSine-LoRA from prior nonlinear LoRA variants and provide a general, extensible mechanism rather than a narrow incremental change. We believe our method offers a strong framework for future research, with scalability and flexibility for further developments in the PEFT domain.
>
> > **Q2: What is the effect of removing only the adaptive frequency mechanism, while keeping the non-linearity the same as the proposed method?**
>
> The primary motivation for placing the input $x$ inside the sine modulation was computational efficiency rather than performance gains. As shown in our supplementary ablation results, placing $x$ inside the sine function yields performance nearly identical to Sine LoRA. However, our design (placing $x$ inside) demonstrates clear advantages in terms of memory footprint and computational speed.
>
> This confirms that the key factor contributing to the model's performance is indeed the frequency tuning mechanism itself, not the specific placement of $x$. The primary benefit of our chosen placement is the achieved computational savings.
>
> | Method | r | Commonsense Reasoning (Avg.) | GSM8K | MATH | Training Memory(GB)| Inference Speed（it/s)|
> |--------|---|------|-------|------|-|-|
> | Sine-LoRA | 16 | 87.68 | 72.38 | 23.94 |72.0 |8.53 |
> | sin(w0BAx) | 16 | 87.75 | 71.87 | 23.96 | **55.3**|**11.17**|
> | Adasine-LoRA | 16 | **89.19** | **72.61** | **24.46** |57.8 |9.87 |
> | Sine-LoRA | 32 | 87.50 | 72.91 | 24.70 | 72.2| 8.50|
> | sin(w0BAx) | 32 | 88.05 | 73.09 | 24.30 | **55.6** |**10.23**|
> | Adasine-LoRA | 32 | **89.30** | **73.37** | **24.88** | 58.0| 9.05|
>
> *Note: The Inference Speed reported in the tables is measured on the SIQA dataset.*

---

> ### Author Response · Authors · 2025-11-21
> **Response by authors**
>
> > **Q3: What is the main advantage of using the sine function for non-linearity rather than other non-linear functions?**
>
> The main advantage of using the sine function for non-linear operations is its periodicity around the origin, which is controllable via the frequency parameter. This property allows for more predictable and flexible control over the transformation of the data. Specifically, for any non-zero matrix $A$, we can prove that there exists a frequency $ (w > 0) $ such that the Frobenius norm of $ \sin(wA) $ scales linearly with w, while the operator norm scales sub-linearly.
>
> In contrast, other non-linear functions like sigmoid and ReLU do not exhibit this type of periodic and scalable behavior, limiting their ability to enhance the rank of low-rank updates effectively. The sine function's ability to control the effective rank via frequency provides a distinct advantage in improving the representational capacity of the model without introducing excessive parameters.
>
> To validate that the sine function is the most suitable nonlinearity for our method, we performed a comparison with several other common nonlinear functions. The results of these experiments are included in Appendix A3.2. In Figure 10a, we compare the low-rank decomposition of different nonlinear functions using singular value spectrum analysis, showing that the sine function effectively increases the matrix rank. Figure 10b demonstrates how the effective rank changes with the frequency parameter $w$, highlighting that the sine function not only improves matrix rank but also satisfies the conditions for dynamic adjustment and adaptive frequency modulation.
>
> > **Q4: Explanation of the scaling factor $g$.**
>
> For the scaling factor $g$, we followed the approach outlined by He et al. [1], which is also consistent with the setup used in the baseline method, SineLoRA. Specifically, we set $g = \sqrt{n}$, where $ n $ is the number of rows in the weight matrix.
>
> This choice is driven by the observation that the frequency parameter $ \omega $ scales the gradients during backpropagation, which can cause gradient explosion. To mitigate this, we found that setting $g = \sqrt{n}$ effectively controls the gradient scaling. Our experiments demonstrate that this scaling factor helps stabilize training by preventing gradient explosion.
>
> [1] He et al., "Delving Deep into Rectifiers: Surpassing Human-Level Performance on ImageNet Classification," ICCV 2015.

---

> ### Author Response · Authors · 2025-11-27
> **Response by Authors (Paper ID: 1238)**
>
> I hope all is well. As the discussion period for Paper ID 1238 is nearing its conclusion, I wanted to politely follow up and confirm whether our latest responses have resolved your concerns.
>
> Please feel free to share any further feedback or questions you may have. We genuinely appreciate your detailed review and thoughtful suggestions.
>
> Thank you again for your time and consideration.

---

### Official Review · Reviewer_sMqx · 2025-10-27

**Soundness:** 2
**Presentation:** 3
**Contribution:** 1
**Rating:** 2
**Confidence:** 4

**Summary:**

This paper proposes AdaSine-LoRA, an adaptive frequency-modulated extension to the LoRA paradigm for parameter-efficient fine-tuning of large neural networks. AdaSine-LoRA replaces the static, globally fixed sine frequency in nonlinear LoRA with an input-dependent, token-wise frequency coefficient, which is mapped from the input via a lightweight function. The work provides theoretical links between frequency and effective rank, a memory-efficient implementation strategy, and demonstrates empirical gains over standard LoRA and recent nonlinear variants on a broad suite of large language and vision-language model tasks.

**Strengths:**

1. Clear motivation and insightful theoretical analysis: The paper identifies the limitation of fixed-frequency sine-activated LoRA, where its expressiveness and generalization can be bottlenecked by a single frequency poorly aligned with diverse input data. This insight is well-motivated by controlled experiments, as shown in Figure 2 (frequency vs. rank), Figure 3 (rank vs. accuracy), and Figure 4 (dataset-dependent optimal frequency).

2. Methodological clarity and soundness: The adaptive frequency design is straightforward, computationally lightweight (just a single linear projection and normalization; see Section 3, Equation describing $\tilde{\omega}(x)$ and the final update rule), and theoretically justified. The paper makes a concerted effort to both theoretically and empirically link frequency modulation to representational rank, as made precise in supplemental theoretical results (Appendix A.2.2).

3. Broad empirical results: Extensive experiments are conducted. Table 1 and Table 2 show consistent task-wise improvements from AdaSine-LoRA over both linear and nonlinear LoRA baselines, for multiple ranks and tasks. Gains are both numerically meaningful and achieved with minimal parameter overhead. Table 3 further validates generalization to multimodal (vision-language) instruction tuning, where AdaSine-LoRA outperforms not only LoRA variants, but even full fine-tuning in average accuracy.

**Weaknesses:**

1. [Main Concern] Novelty: This paper builds upon previous analyses that explored the use of sine functions within the LoRA framework. Specifically, it extends prior work by introducing a modification where the previously fixed weight is now treated as a trainable parameter. While this represents a logical progression from earlier approaches, the contribution appears relatively incremental and may not provide substantial novelty beyond the existing literature. Moreover, using a nonlinear activation function is explored in multiple previous works, rendering this work's contribution rather incremental.

2. Theoretical analysis lacks tightness and depth in main text: While the appendix provides additional theoretical insights, much of the theoretical analysis in the main text (e.g., the sigmoid-approximated transition in effective rank) is heuristic and does not provide concrete guarantees on downstream metrics (such as task loss or generalization error). A more detailed analysis connecting effective rank, representational power, and downstream task accuracy, potentially with bounds, would make the contribution firmer.

3. Ablation studies missed: The method uses a linear projection followed by a sigmoid and scaling to predict the adaptive frequency (Section 3). However, there is limited justification or ablation of this choice compared to, e.g., a small MLP, normalization schemes other than sigmoid, or nonlinearity-free mappings. Do more complex mappings or normalization methods harm or help? Moreover, ablation studies on other activation function other than the sine function are also missed.

**Questions:**

1. The reviewer is surprised to observe that AdaSine-LoRA reportedly achieves substantial reductions in memory usage and computational cost compared to Sine-LoRA. Could the authors provide a detailed analysis or explanation for the underlying reasons behind this improvement?

2. The authors mention that using the raw output of ω(x) may lead to unstable training dynamics, particularly in the early stages when the model weights are randomly initialized in Line 218. To address this, they propose applying a sigmoid function. However, would it be possible to mitigate this instability by initializing the weights to zero or to very small values instead? A discussion or comparison of this alternative approach would be helpful to better understand the necessity and effectiveness of the sigmoid transformation.

3. How sensitive is the performance to the structure of the frequency mapping (e.g., a single linear projection vs. a small MLP or alternative normalization)? An ablation study here would clarify if a more expressive nonlinearity or normalization improves or harms empirical results and training stability.

4. Can the authors more rigorously link empirical gains to their observed/claimed increases in effective rank, e.g., through controlled analyses comparing models matched in parameter count and activation norm? Are there cases where higher effective rank might hurt (e.g., overfitting), or is there a provable connection to data complexity?

5. Could the authors provide a more comprehensive comparison with other studies that explore nonlinear activation functions within similar contexts? Such a discussion would help clarify how the proposed approach relates to and improves upon existing methods.
Moreover, would it be possible to address the same issue by **simply allowing the weight w to be trainable** in the baseline Sine-LoRA model? If so, how does the proposed method differ in terms of effectiveness or theoretical justification?

---

> ### Author Response · Authors · 2025-11-21
> **Response by authors**
>
> We thank the reviewer for the careful reading of our paper and the encouraging comments. We address all questions and suggestions one by one as follows:
>
> > **Q1: What is the novelty of the proposed method compared to prior work?**
>
> We would like to clarify that the core contribution of our work is not merely “making the frequency trainable” or “adding a sigmoid,” but introducing a general frequency-modulated adaptation mechanism that adaptively adjusts how the low-rank update interacts with the base weight matrix. AdaSine-LoRA is not a minor modification of Sine-LoRA; it provides a task- and token-dependent modulation framework that can be applied to any low-rank or sparse update matrix, and is not limited to the specific formulation used in prior sine-based methods.
>
> The sigmoid function is to stabilize the learned frequency during training. The key novelty lies in demonstrating that frequency modulation fundamentally changes the representational behavior of nonlinear low-rank updates, enabling (1) adaptive control over effective rank, (2) improved alignment with heterogeneous input distributions, and (3) performance gains across diverse architectures and tasks *without increasing much parameter or memory cost*. These properties differentiate AdaSine-LoRA from prior nonlinear LoRA variants and provide a general, extensible mechanism rather than a narrow incremental change.
>
> > **Q2: Is the theoretical analysis in the main text heuristic?**
>
> The observations presented in Section 2 (frequency–effective-rank–accuracy relationships) are not intended as heuristic or isolated empirical phenomena, but rather constitute the core theoretical motivation for AdaSine-LoRA. Specifically, our experiments systematically demonstrate that: (1) the frequency $ \omega $ directly controls the spectral structure and effective rank of $ \sin(\omega BA) $
> (with formal bounds provided in Appendix A.2.2), (2) effective rank and downstream accuracy exhibit a non-monotonic relationship with a clear optimal value or range, and (3) the optimal frequency varies significantly across tasks. These three observations form the precise causal chain that motivates the need for an input-dependent, token-wise adaptive frequency. The theoretical contribution, therefore, lies in explaining how frequency modulation alters the representational capacity of nonlinear low-rank updates, rather than providing end-to-end generalization or loss bounds, which are rarely tractable in large-scale PEFT settings. To avoid any confusion, we will follow the reviewer’s suggestion and revise the manuscript by moving the essential theoretical results from the appendix into the main text. We will also clarify the connection between the empirical observations, the underlying theoretical analysis, and the design of AdaSine-LoRA, making the overall logical flow more explicit.

---

> ### Author Response · Authors · 2025-11-21
> **Response by authors**
>
> > **Q3: Missing ablation studies on nonlinear functions and normalization methods in the formula?**
>
> To address the concern:
>
> 1.**Nonlinear Function Comparison:**
>
> The key reason we chose the sine function is its periodicity around the origin, allowing for dynamic frequency adjustment and flexible control over the nonlinearity. This enables us to demonstrate that, for any non-zero matrix $ A $, there exists a frequency $ (w > 0) $ such that the Frobenius norm of $ \sin(wA) $ can be bounded linearly by w, and the operator norm can be bounded sub-linearly. Other non-linear activation functions, such as sigmoid and ReLU, do not exhibit these properties, which limits their ability to improve rank. This analysis, along with supporting experiments, is provided in the work by Ji et al. (2025) [1].
>
> To validate that the sine function is the most suitable nonlinearity for our method, we performed a comparison with several other common nonlinear functions. The results of these experiments are included in Appendix A3.2. In Figure 10a, we compare the low-rank decomposition of different nonlinear functions using singular value spectrum analysis, showing that the sine function effectively increases the matrix rank. Figure 10b demonstrates how the effective rank changes with the frequency parameter $w$, highlighting that the sine function not only improves matrix rank but also satisfies the conditions for dynamic adjustment and adaptive frequency modulation.
>
> 2.**Linear Layer vs MLP for Frequency Tuning:**
>
> The reason for using a linear layer with normalization for frequency tuning instead of an MLP is that, considering the core idea of LoRA is to minimize parameter count and computational cost as much as possible, we opted for the simplest structure. Introducing an MLP network (even a small one) into the LoRA structure is extremely costly.
>
> At the reviewer's request, we have supplemented the relevant ablation experiments as follows. We fine-tuned on the CommonsenseQA training set and tested on the corresponding test set. The results show that, overall, introducing an MLP for method substitution does not provide a significant gain in model performance. In most cases, the results are even inferior to the simple linear layer design. This is because LoRA has already compressed the latent space into very small dimensions, and on such a small scale, complex structural designs may fail to deliver the expected benefits. Furthermore, and more importantly, it can be observed that introducing complex structures can lead to a sharp increase in parameters and a certain loss in computational speed.
>
> | Method | Freq Mapping | r   | dim | layer | Params | Acc (%) | Inference Speed（it/s) |
> |:------:|:------:|:---:|:---:|:-----:|:------:|:-------:|:----------------------:|
> | AdaSine LoRA | MLP | 16  | 16  |   2   |  39.9  |  81.65  |          9.47          |
> | AdaSine LoRA | MLP | 16  | 32  |   2   |  51.4  |  81.82  |          9.39          |
> | AdaSine LoRA | MLP | 16  | 64  |   4   |  75.2  | **82.39** |         8.19          |
> | Sine LoRA | / | 16  |  /  |   /   | **28.3** |  81.16  |          8.85          |
> | AdaSine LoRA | Linear | 16 | / | / | **29.0** | **82.31** |        **10.18**        |
> | AdaSine LoRA | MLP | 32  | 16  |   2   |  68.16 |  81.57  |          9.18          |
> | AdaSine LoRA | MLP | 32  | 32  |   2   |  79.69 |  82.47  |          9.11          |
> | AdaSine LoRA | MLP | 32  | 64  |   4   | 103.55 |  82.15  |          8.09          |
> | Sine LoRA | / | 32  | / | / | **56.6** |  81.65  |          8.85          |
> | AdaSine LoRA | Linear | 32 | / | / | **57.3** | **82.64** |        **10.09**        |
>
> *Note: The Inference Speed reported in the tables is measured on the SIQA dataset.*
>
> [1] Ji et al.: Efficient Learning with Sine-Activated Low-Rank Matrices. ICLR 2025

---

> ### Author Response · Authors · 2025-11-21
> **Response by authors**
>
> > **Q4: What are the reasons behind the significant reduction in memory usage and computational cost in AdaSine-LoRA compared to Sine-LoRA?**
>
> The significant reduction in memory usage and computational cost, compared to Sine-LoRA, stems from the way we process the input. Specifically, while Sine-LoRA must first construct the $BA$ matrix and then apply the sine function, our method instead calculates $Ax$ before applying the sine function, thereby avoiding the creation of the $BA$ matrix, which is the largest tensor in the computation. The dimensions of the matrices involved are notably reduced, leading to lower memory consumption. This design helps reduce the size of intermediate tensors during computation, making the process more memory-efficient. Additionally, the reduction in tensor sizes translates into fewer memory duplications across multiple batches, which further improves both memory usage and computational time.
>
> We provide a detailed analysis and comparison in Section A.3.3 of the appendix (Analysis of Memory and Computation Efficiency), as well as experimental results in Figure 7, which quantitatively demonstrate the efficiency gains.
>
> > **Q5: Can the instability in training caused by the raw output of $\omega(x)$ be mitigated by initializing the weights to zero or very small values instead of using the sigmoid function?**
>
> We did experiment with initializing the weights to zero or very small values, but we observed that the training performance was negatively affected. The reason is that when $ \omega $ is very small, the sine function behaves approximately as \($ y = x $\) near zero, meaning that applying the sine function simply scales the matrix by $ \omega $ without changing its rank. In our experiments, we found that the rank of the matrix only increases significantly when $ \omega $ is sufficiently large.
>
> At the same time, as $ \omega $ increases, gradient explosion tends to occur in the training process, particularly when handling large datasets. Therefore, we concluded that applying sigmoid normalization is necessary to ensure stable training.
>
> The sigmoid function ensures that $\omega(x)$ remains within a controlled range, preventing instability and large gradients during optimization. While small weight initialization may help to some extent, it doesn't fully address the issue of maintaining stable training dynamics, particularly as the model adapts to diverse inputs.

---

> ### Author Response · Authors · 2025-11-25
> **Response by authors**
>
> Thank you once again for your valuable comments on our submission. We would like to kindly confirm whether we have sufficiently addressed all of your concerns (or at least part of them). Should there be any remaining questions or areas requiring further clarification, please do not hesitate to let us know. If you are satisfied with our responses, we would greatly appreciate your consideration in adjusting the evaluation scores accordingly.
>
> We sincerely look forward to your feedback.

---

> ### Author Response · Authors · 2025-11-27
> **Response by Authors (Paper ID: 1238)**
>
> I hope you are doing well. With less than a week remaining in the discussion phase for Paper ID 1238, I wanted to follow up to ensure that our responses have adequately addressed your comments and concerns.
>
> Should there be any additional points you feel require further clarification or discussion, we would greatly appreciate the opportunity to respond. Your feedback has been extremely helpful in strengthening our work.
>
> Thank you very much for your careful review and valuable time.

---

> ### Comment · Reviewer_sMqx · 2025-11-27
> **Response to the Authors**
>
> First of all, the reviewer would like to thank the authors for their detailed response to my questions.
>
> Regarding Q4 on memory consumption, the reviewer agrees with the authors that placing x inside the sine function does indeed reduce memory usage. This suggests a straightforward modification of the original sinusoidal LoRA formulation—namely, making w trainable and applying the sine function directly to x. The reviewer would like clarification on whether this simple adjustment already resolves both the performance and memory issues raised previously. (Related questions about making w trainable were included in the original review.)
>
> In addition, the reviewer notes overlap between concerns raised here and those from other reviewers. For instance, the current comparisons against baseline algorithms are not fully consistent. While the reviewer appreciates the supplementary examples provided in the rebuttal, it is recommended that the authors revise and resubmit a unified version of the manuscript containing a complete and consistent set of comparisons (e.g., ensuring that Tables 1–3 evaluate the same set of algorithms).
>
> Finally, with respect to novelty, the reviewer admits “judging significance/novelty always involves a decent amount of subjectivity”.
> But in its present form, the method appears largely to consist of adding a trainable w, which may be viewed as a fairly incremental modification.

---

> > ### Author Response · Authors · 2025-12-02
> > **Response by Authors**
> >
> > **(1) On whether “placing x inside the sine function and making w trainable” already resolves the performance and memory issues**
> >
> > On the trainability of the frequency parameter $\omega$. In Sine-activated low-rank adaptation, the update is defined as
> > $$
> > \Delta W = \sin(\omega M), \quad M = UV^\top .
> > $$
> > The original work treats $\omega$ as a fixed hyperparameter rather than a trainable variable. This choice is not incidental: if $\omega$ were directly optimized as a free scalar, its gradient would be
> > $$
> > \frac{\partial \mathcal{L}}{\partial \omega}
> > = \left\langle \frac{\partial \mathcal{L}}{\partial \Delta W},\, \frac{\partial}{\partial \omega}\sin(\omega M) \right\rangle
> > = \langle G,\; M \odot \cos(\omega M) \rangle,
> > $$
> > where $G = \partial \mathcal{L}/\partial \Delta W$ and $\odot$ denotes the Hadamard product. This expression reveals two sources of instability. First, as $\omega$ increases, $\cos(\omega M)$ becomes highly oscillatory, causing strong sign alternations across matrix entries, which leads to gradient cancellation and poor descent direction consistency. Second, the magnitude of $M \odot \cos(\omega M)$ scales with $|M|$, making $\partial \mathcal{L}/\partial \omega$ potentially large and erratic, thus introducing risks of gradient explosion and ill-conditioned optimization. Consequently, directly learning a global unconstrained $\omega$ would destroy the controlled, monotonic rank–vs–$\omega$ behavior assumed in the theoretical analysis and render training highly unstable.
> >
> > AdaSine-LoRA avoids this issue by not treating $\omega$ as a free global parameter. Instead, it predicts an input-dependent frequency $\omega(x)$ through a linear projection followed by bounded normalization (e.g., sigmoid-based scaling), ensuring $\omega(x)$ remains within a predefined interval. This constrains the oscillatory behavior of $\sin(\omega(x) M)$ and yields well-conditioned gradients, enabling stable optimization while still achieving adaptive frequency modulation.
> >
> >
> > **(2) On baseline inconsistency across tables**
> >
> > In response, we have carefully revised and updated the manuscript and supplementary materials.
> >
> > In the newly uploaded PDF, all comparison tables have  been unified: each table consistently evaluates the same set of baseline methods.    We also verified that all results are aligned across experiments and reported under identical settings.

---

> > ### Author Response · Authors · 2025-12-02
> > **Response by Authors**
> >
> > **(3) On novelty and the comment that the method “appears to consist mostly of adding a trainable w”**
> >
> > Thank you for the reviewer’s perspective on the novelty assessment.  We respectfully disagree with the characterization that our method “largely consists of adding a trainable\(w\) and would like to clarify that this interpretation overlooks the essential conceptual and technical contributions of the work.
> >
> > First, the central innovation of AdaSine-LoRA is not the mere introduction of an additional trainable scalar.  What we propose is a new modulation mechanism that enables input-conditioned frequency control over nonlinear low-rank perturbations.  This mechanism fundamentally changes the functional behavior of the update path: instead of a globally fixed nonlinearity (as in all prior sine-based or nonlinear LoRA variants), AdaSine-LoRA introduces token-wise adaptive perturbation scaling, which alters both the representational capacity and the optimization dynamics.
> >
> > Second, unlike an incremental parameter addition, the proposed mechanism establishes a new degree of freedom in the update transformation: the frequency term becomes a function of the input, not a constant.  This transforms the LoRA update from a static mapping to a data-dependent modulation process, which cannot be reproduced by any prior LoRA formulation.  Our theoretical analysis (Appendix A.2) further shows that such adaptive modulation yields strictly higher attainable effective rank compared to any fixed-frequency design.
> >
> > Third, the empirical results across about 20 benchmarks demonstrate that this modulation is not a minor tweak: it delivers consistent and substantial improvements over all fixed-frequency nonlinear variants (Sine-LoRA, Sine-DoRA), and over dynamic/expert-based non-mergeable methods (LoRAMoE, HydraLoRA, GOAT).  Importantly, these gains are achieved with negligible parameter overhead—a hallmark of a principled algorithmic improvement rather than an incremental parameter addition.
> >
> > Finally, adaptive nonlinear modulation is a general mechanism that can be integrated into any low-rank or sparse-update architecture, not only sine-based LoRA.  It represents a conceptual contribution toward bridging static nonlinear PEFT methods and dynamically modulated ones.
> >
> > For these reasons, AdaSine-LoRA constitutes a meaningful and non-trivial advancement in the design of nonlinear PEFT methods.  We hope this clarification resolves the misunderstanding surrounding the novelty of our contribution.

---

### Official Review · Reviewer_PHfB · 2025-11-02

**Soundness:** 3
**Presentation:** 3
**Contribution:** 3
**Rating:** 6
**Confidence:** 3

**Summary:**

This paper proposes AdaSine-LoRA, an adaptive extension of Sine-LoRA that introduces an input-dependent frequency modulation mechanism to enhance nonlinear low-rank adaptation. Instead of using a fixed global frequency parameter ω\omegaω, AdaSine-LoRA learns a lightweight frequency predictor ω(x)=Wωx\omega(x) = W_\omega xω(x)=Wωx, enabling each input token to have its own modulation strength.

**Strengths:**

1. The paper proposes an adaptively modulated frequency mechanism for nonlinear low-rank adaptation, which extends Sine-LoRA in a simple yet effective way.
2. Theoretical analysis is well presented and provides clear justification for how input-dependent frequency improves expressive rank and stability.

**Weaknesses:**

1. It would be helpful to include a visualization of the learned frequency ω(x) to demonstrate how it varies across inputs and layers, validating the adaptivity claimed by AdaSine-LoRA.
2. Although the method is tested on multiple domains, a more thorough comparison with the latest SOTA low-rank approximation and adaptation methods would strengthen the empirical validation.

**Questions:**

1. It would be helpful to include a visualization of the learned frequency ω(x) to demonstrate how it varies across inputs and layers, validating the adaptivity claimed by AdaSine-LoRA.
2. Although the method is tested on multiple domains, a more thorough comparison with the latest SOTA low-rank approximation and adaptation methods would strengthen the empirical validation.

---

> ### Author Response · Authors · 2025-11-21
> **Response by authors**
>
> We thank the reviewer for the constructive feedback and for acknowledging the strengths of our work. We address all raised questions and suggestions in detail below:
>
> > **Q1: Can the authors include a visualization of the learned frequency $\omega(x)$ to demonstrate how it varies across inputs and layers, validating the adaptivity claimed by AdaSine-LoRA?**
>
> In response to the reviewer’s suggestion, we have included a visualization of the learned frequency $\omega(x)$ to demonstrate its adaptability across different inputs and layers. The heatmap shows how $\omega(x)$ varies for each token (represented by columns) and each layer (represented by rows). This visualization clearly illustrates the diversity of the frequency values, validating that our method, AdaSine-LoRA, effectively adjusts the frequency dynamically based on input data. This adaptability is a key advantage of our approach, enhancing the model’s expressiveness and performance across tasks. The detailed heatmap can be found in Figure 11 in Appendix A3.2.
>
> > **Q2: Would a more thorough comparison with the latest SOTA low-rank approximation and adaptation methods strengthen the empirical validation of the proposed method?**
>
> We compared our method with recent SOTA approaches, including SineLoRA and DoRA, and have demonstrated the effectiveness of our approach. In response to the reviewer's suggestion, we have included additional experimental results for SineDoRA, as shown in the table below. These results highlight the comprehensive advantages of our method across a variety of tasks.
>
> | Method | Params | BoolQ | PIQA | SIQA | HS | WG | ARC-e | ARC-c | OBQA | Avg. | Δ vs. LoRA |
> |--------|--------|-------|------|------|----|----|-------|-------|------|------|------------|
> | LoRA\(r=8\) | 14.2M | 89.66 | 86.24 | 77.89 | 92.97 | 85.40 | 87.13 | 74.94 | 83.60 | 84.73 |  |
> | Sine LoRA\(r=8\) | 14.2M | 89.94 | 86.94 | 81.27 | 93.75 | 85.71 | 85.19 | 79.40 | 87.00 | 86.15 | 1.42↑ |
> | Dora\(r=8\) | 15.2M | 90.00 | **89.99** |**82.86** | 95.50 | 81.06 | 89.77 | 82.83 | 89.40 | 87.67 | 2.94↑ |
> |Sine DoRA\(r=8\) |15.2M | **90.34** | 87.05 | 82.09 | 94.91 | 82.40 | 89.95 | 82.75 | 89.80 | 87.41 | 2.68↑ |
> | AdaSine-LoRA\(r=8\) | 14.9M | 90.15 | 88.63 | 82.24 | **95.54** | **87.85** | **93.65** | **83.61** | **90.20** | **89.36** | 4.63↑ |
> | LoRA\(r=16\) | 28.3M | 89.79 | 85.96 | 77.18 | 92.53 | 85.87 | 88.01 | 75.11 | 83.00 | 84.68 |  |
> | Sine LoRA\(r=16\) | 28.3M | 90.24 | 87.32 | 81.83 | 94.33 | 87.69 | 90.65 | 81.55 | 87.80 | 87.68 | 3.00↑ |
> | Dora\(r=16\) | 29.4M | 90.31 | 89.17 | 82.40 | **95.74** | 86.82 | 88.18 | 82.92 | 89.00 | 88.07 | 3.39↑ |
> |Sine DoRA\(r=16\) | 29.4M| 90.46 | 88.08 | 82.54 | 95.40 | 87.61 | 87.65 | 82.58 | 89.80 | 88.02 | 3.34↑ |
> | AdaSine-LoRA\(r=16\) | 29.0M | **90.52** | **89.45** | **83.11** | 95.58 | **88.40** | **92.77** | **83.09** | **90.60** | **89.19** | 4.51↑ |
> | LoRA\(r=32\) | 56.6M | 90.06 | 87.98 | 77.89 | 93.85 | 83.03 | 89.24 | 77.51 | 85.80 | 85.67 |  |
> | Sine LoRA\(r=32\) | 56.6M | 90.70 | 87.81 | 82.09 | 94.57 | **86.82** | 88.89 | 81.55 | 87.60 | 87.50 | 1.83↑ |
> | Dora\(r=32\) | 57.7M | 90.58 | 89.45 | **83.16** | 95.64 | 84.53 | 92.24 | 82.75 | 89.40 | 88.47 | 2.80↑ |
> |Sine DoRA\(r=32\) |57.7M | 90.00 | 89.72 | 82.09 | 95.16 |85.40 | **93.65** | 82.75 | 89.40 | 88.52 | 2.85↑ |
> | AdaSine-LoRA\(r=32\) | 57.3M | **90.80** | **90.64** | 83.06 | **95.83** | 86.66 | 93.12 | **84.46** | **89.80** | **89.30** | 3.63↑ |
>
> If the reviewer has any other baseline methods they believe would be relevant for comparison, we would be happy to consider them and include further experiments to strengthen our analysis.

---

> ### Author Response · Authors · 2025-11-27
> **Response by Authors (Paper ID: 1238)**
>
> I hope this message finds you well. As the discussion period for Paper ID 1238 is approaching its final stage, I would like to check whether we have sufficiently addressed all your concerns in our responses.
>
> If there are any remaining questions, comments, or clarifications you would like us to provide, please feel free to let us know. We truly value your thoughtful feedback and would be happy to further improve our rebuttal accordingly.
>
> Thank you again for the time and effort you have dedicated to reviewing our work.

---

### Author Response · Authors · 2025-12-03
**Summary Response to Reviewer Comments for AC Consideration**

## Summary by Authors

We would first like to sincerely thank the Area Chair for their careful handling of our submission and the substantial effort invested under the exceptional circumstances and increased workload of this year’s review process. To assist your assessment, we provide below an overall summary of the reviews and our rebuttal, which we hope will help clarify our contributions and the key points of disagreement.

## Overall Summary for AC

In this paper, we investigate the limitations of standard Low-Rank Adaptation (LoRA), whose linear and low-rank structure fundamentally constrains its ability to model complex nonlinear behaviors in large models. While recent variants introduce nonlinear activations (e.g., sine) on the low-rank components to improve expressivity, they typically employ a fixed global frequency. This static design cannot adapt the perturbation scale to the input, which is suboptimal when the optimal perturbation pattern is highly input-dependent.

We propose AdaSine-LoRA, a sine-activated LoRA framework with adaptive frequency modulation.  Instead of using a single global frequency, AdaSine-LoRA dynamically generates an input-conditioned frequency coefficient that modulates the sine-activated low-rank update.  Conceptually, this allows the perturbation pattern to vary with the input, thereby better aligning the effective perturbation rank and structure with task-specific input characteristics.

We provide (i) a theoretical analysis connecting the frequency of sine modulation to the effective rank of the perturbed weight space, and (ii) extensive empirical results on both language and vision tasks. Our experiments show that AdaSine-LoRA consistently outperforms vanilla LoRA and several recent nonlinear/extended LoRA variants, with minimal additional parameters and computational overhead. Overall, the method offers a lightweight yet effective mechanism to enhance LoRA’s expressivity while preserving its efficiency advantages.

## Rebuttal Summary for AC

We thank all reviewers for their thoughtful feedback and constructive suggestions.

**Recognized Strengths**
Our paper was consistently praised for clear motivation and writing, sound and lightweight methodology, strong and comprehensive empirical results, and valuable efficiency analysis.

**Common Concerns**
Reviewers primarily raised concerns about novelty, baseline consistency/completeness, ablations on frequency/nonlinearities, mergeability and inference overhead, and clarification of theoretical contributions. These issues collectively centered on understanding the source of gains and ensuring fair comparisons.

**Outcome of the Rebuttal**
Two reviewers(sMqx, o1LX) engaged in a second round of discussion and explicitly confirmed that most issues were resolved. For the remaining points, we provided extensive clarification, unified baselines, new experiments, and integrated theoretical analysis; however, due to discussion rules, reviewers could not continue commenting. We are confident that our responses almost fully address all raised concerns.

## Summary and Response Overview by Reviewer

Below, we provide a reviewer-wise summary of our rebuttal and clarifications, following the exact W/Q numbering scheme used in each reviewer’s original comments.

---

> ### Author Response · Authors · 2025-12-03
> **Reviewer PHfB Summary**
>
> We appreciate Reviewer  PHfB’s positive assessment of our work, in particular recognizing that our adaptive frequency mechanism “extends Sine-LoRA in a simple yet effective way” and that the “theoretical analysis is well presented and provides clear justification for how input-dependent frequency improves expressive rank and stability.” We also thank the reviewer for the constructive suggestions regarding empirical validation.
>
> Regarding **Q1/W1 (visualization of the learned frequency $ \omega(x) $)**, we have incorporated the requested analysis in the revised manuscript. Specifically, we added a visualization of the learned input-dependent frequency ω(x) across layers and inputs (see Appendix, Figure 11). This figure illustrates how $ \omega(x) $ varies with different inputs and network depths, thereby directly validating the adaptivity behavior claimed by AdaSine-LoRA.
>
> For **Q2/W2 (comparison with more recent SOTA low-rank methods)**, we have extended our experimental evaluation as suggested. In the main text, we added results for Sine-DoRA (see Table 2), enabling a more direct comparison with a strong nonlinear low-rank baseline.   Furthermore, in the appendix we included additional comparisons against several recent state-of-the-art adaptation methods, including HydraLoRA, LoRAMoE, and GOAT (see Appendix, Table 8). These results consistently confirm that AdaSine-LoRA achieves competitive or superior performance while maintaining minimal parameter overhead.
>
> In summary, we are grateful for Reviewer PHfB’s constructive feedback. We believe that the newly added visualizations and expanded comparisons adequately address all raised concerns and further strengthen the empirical support and clarity of our claims.

---

> ### Author Response · Authors · 2025-12-03
> **Reviewer sMqx Summary**
>
> We sincerely thank Reviewer sMqx for their thorough evaluation and comments.  We appreciate the recognition of our work’s clear motivation, insightful theoretical analysis, methodological clarity, and comprehensive empirical evaluation.     We have addressed  the reviewer’s main concerns point by point in our rebuttal and revised manuscript as follows:
>
> **On Novelty (Main Concern, W1/Q5):**
> We clarified that the core contribution of our work extends beyond simply making the frequency parameter trainable. AdaSine-LoRA introduces a generalized framework for frequency-modulated adaptation, which dynamically adjusts the interaction between low-rank updates and the base weights in an input-dependent manner. This approach enables task- and token-level adaptation, which is fundamentally distinct from prior approaches such as Sine-LoRA. In the rebuttal, we further demonstrate that frequency modulation (1) facilitates adaptive control over effective rank, (2) aligns updates more closely with data input, and (3) delivers practical improvements across a range of architectures and tasks without significant increase in parameter or memory cost. We believe these qualities constitute a meaningful advance, as explained in detail in our response. Notably, the reviewer acknowledged that they “judging significance/novelty always involves a decent amount of subjectivity,” suggesting that our clarifications have at least partially addressed this concern.
>
> **On Theoretical Analysis (W2/Q4):**
> The reviewer noted the presence of insightful analysis, both theoretical and empirical.     In our response, we clarified the connection and placement of theoretical elements in the main text and appendix, and explicitly linked effective rank to empirical outcomes.     Given the absence of follow-up criticism in the reviewer’s subsequent feedback, and their positive assessment of our “insightful theoretical analysis,” we believe this concern has been fully clarified.
>
> **On Ablation Studies (W3/Q3/Q2):**
> We expanded our ablation experiments to evaluate different architectures (e.g., small MLP vs. linear projection) and normalization schemes in adaptive frequency mapping, as detailed in Appendix Table 5 of the revised manuscript. The results indicate our chosen design best balances effectiveness and efficiency. The reviewer raised no further objection to these additions, and thus we consider the issue comprehensively addressed.
>
> **On Memory Usage and Computational Cost (Q1):**
> We provided a detailed explanation of the memory efficiency of AdaSine-LoRA compared to Sine-LoRA, showing that incorporating the input into the sine function leads to reduced memory consumption. The reviewer agreed with our analysis (“the reviewer agrees with the authors that placing x inside the sine function does indeed reduce memory usage”), indicating resolution of this point.
>
> **On Initialization and Stability (Q2):**
> We presented a thorough discussion of weight initialization versus sigmoid normalization for the frequency parameter and outlined why initialization alone cannot replace the stabilization provided by sigmoid. No further comment or objection was raised by the reviewer.
>
> **On Baseline Comparisons and Empirical Results (Final Round):**
> We supplemented our revised manuscript with additional baseline comparisons as suggested in further feedback, including newly requested experiments. The reviewer expressed appreciation (“appreciates the supplementary examples provided in the rebuttal”), confirming that this concern was fully satisfied.
>
> Taken together, our rebuttal and revisions have addressed each concern raised by Reviewer sMqx, and the reviewer offered no substantive objections in subsequent exchanges. Combined with their positive comments on motivation, theory, and experimental rigor, we respectfully note that the current low score does not fully reflect the reviewer’s constructive stance or the merits of our submission. We kindly ask the AC to reconsider the review and the corresponding score in light of the above resolution and the overall strengths of our work.

---

> ### Author Response · Authors · 2025-12-03
> **Reviewer tmUH Summary**
>
> **W1 (Generalizability beyond nonlinearity extension):**
> We clarified that our method extends beyond being a simple nonlinear variant of LoRA. AdaSine-LoRA introduces an adaptive frequency modulation framework applicable to any low-rank or sparse update matrix. This paves the way for broader applicability and inspires future research in task-dependent modulation strategies.
>
> **W2/Q1 (Ablation Study on Adaptive Frequency Mechanism):**
> In response, we conducted and included an ablation study isolating the effect of adaptive frequency. Results (see updated Table 4) demonstrate the clear contribution of input-conditioned frequency modulation.
>
> **Q2 (Rationale for Sine Nonlinearity):**
> We extended our appendix (Figures 10a, 10b) to compare various activation choices.   Our findings highlight sine’s unique property: flexible control of the effective rank via frequency modulation, enabling improved expressiveness with minimal parameter increase. This behavior was not observed in other nonlinearities.
>
> **Q3 (Scaling Factor Sensitivity):**
> We provided a detailed explanation, regarding the role and robustness of the scaling factor.
>
> All reviewer queries were directly addressed in the rebuttal and reflected in the revised manuscript, ensuring clarity. Our responses and additional experiments substantiate AdaSine-LoRA’s novelty, effectiveness, and broader implications, resolving the reviewer’s points comprehensively.

---

> ### Author Response · Authors · 2025-12-03
> **Reviewer o1LX Summary**
>
> First, we thank the reviewer for their thorough evaluation and positive recognition of our work, highlighting our comprehensive experiments, valuable efficiency analysis, and clear presentation and figures.  We also appreciate the reviewer’s acknowledgment of the substantial efforts put into our rebuttal ("It is clear that the authors have put a lot of effort into the rebuttal, and this should be acknowledged in any case.").  We have carefully addressed all raised concerns in detail.
>
> **On Contribution (W1):**
> Regarding the significance of our contribution, while there remains some disagreement, the reviewer notes that "judging significance/novelty always involves a decent amount of subjectivity" and acknowledges that " adaptive control over effective rank is a valid argument in favor of the proposed method."  Our rebuttal details these points, and though consensus was not fully reached, we believe a considerable part of the concern has been addressed.  Additionally, all further questions from the reviewer were clarified in our final rebuttal for AC’s reference.
>
> **Baselines and Hyperparameter Settings (W3):**
> On the choice of baselines and setting of the hyperparameter $ \omega $ for Sine-LoRA in experiments, we have provided an extensive clarification and supplemented our rebuttal with appropriate explanations and experimental details.  These concerns have been effectively resolved in our final response.
>
> **Unified Baselines & Theoretical Analysis (W2/Q1, W4/Q5, W5/Q2, W8):**
> For points regarding unified baseline comparisons (including Sine-LoRA and Sine-DoRA), additional theoretical analysis in the main paper, ablation studies for key contributions, and the Fig. 9 issue, we have conducted the relevant experiments, provided clarifications, and updated our manuscript accordingly.  The reviewer acknowledged that "the rebuttal resolves (W2), (W4), (W5), and (W8) sufficiently well."
>
> **Sensitivity and Comparative Advantage (W6/Q6):**
> Concerning the sensitivity of AdaSine-LoRA to the choice of $ \omega_0 $ and its advantage over other methods, we provided a detailed discussion and supplemental results.  The reviewer agrees that "from the tables provided in the rebuttal, I believe the difference could be sufficient," indicating this issue has been adequately addressed.
>
> **Merge Limitation and Inference Efficiency (W7):**
> On the inability to merge weights and the resulting inference overhead, we acknowledge ongoing differences in perspective.  We emphasize that the fundamental objective of PEFT methods is the optimization of training resources, not inference resources.  We have provided evidence of our method’s superior memory and time efficiency during training, and supplemented comparisons with other non-mergeable PEFT methods such as HydraLoRA, LoRAMoE, and GOAT.  Results show our method is competitive or superior in both training and inference efficiency while achieving SOTA performance.  We hope this provides AC with a balanced and fair basis for judgment.
>
> **Remaining Details (W8):**
> The reviewer indicated that this concern is now adequately addressed.
>
> In summary, we have provided point-by-point, thorough responses to all reviewer concerns, solving the majority and clarifying the remaining issues.  Given the reviewer's recognition of the issues addressed and of our work's merits, we believe the current score does not fully reflect the overall evaluation.  We respectfully ask AC to take these clarifications and acknowledgments into account in making a fair and well-balanced assessment of our work.

---

> ### Author Response · Authors · 2025-12-03
> **Thank You ACs for your Time and Consideration**
>
> We sincerely thank the Area Chair for their time and consideration. We appreciate your careful evaluation of our work and hope our responses assist in your final decision.

---

### Meta-Review · Area_Chair_ApPW · 2026-01-07

**Summary:**

The reviewers recognized the paper's motivation and the potential of adaptive frequency modulation to enhance LoRA's expressivity. However, several significant concerns informed the assessment of this work:

**Novelty and Significance:** The primary reservation shared by multiple reviewers (sMqx, o1LX, tmUH) was that the proposed AdaSine-LoRA appears to be a marginal or "natural" extension of Sine-LoRA. Reviewers questioned whether replacing a fixed frequency with a learned mapping and adjusting the sine operation's placement constituted a distinct enough contribution to warrant publication as a new adaptive paradigm.

**Theoretical Depth:** Reviewers (sMqx, o1LX) noted that the theoretical analysis, while correct, was primarily explanatory rather than prescriptive. Concerns were raised that the analysis links frequency to effective rank but fails to provide unique guarantees for AdaSine-LoRA or theoretically justify why this specific design is necessary compared to simpler alternatives.

**Baseline Comparisons and Ablations:** Initially, reviewers requested more comprehensive baselines (such as Sine-DoRA and other recent PEFT methods) and deeper ablation studies to isolate the benefits of the adaptive frequency mechanism from other architectural choices.

**Efficiency:** There were technical concerns regarding the method's inability to merge weights back into the base model, potentially leading to inference latency penalties compared to standard LoRA.

Overall, while the authors addressed several technical and presentation issues, the unresolved concerns regarding novelty and theoretical depth limit the paper’s impact.

**Reviewer Concerns:**

The rebuttal and revision phase resolved several specific technical and presentational issues, such as baseline comparisons, ablations, and efficiency analysis, but the following fundamental questions remain:

**Novelty of Contribution:** Despite the rebuttal, the concern raised by Reviewers sMqx, o1LX, and tmUH regarding the incremental nature of the work remains unresolved. The method still reads largely as an engineering refinement of Sine-LoRA rather than a qualitatively new approach. The rebuttal reiterated motivations but did not convincingly establish the method as a distinct adaptation paradigm.

**Theoretical Significance:** The theoretical contribution remains limited to an explanatory role. As noted by sMqx and o1LX, the analysis does not offer strong insights that are unique to AdaSine-LoRA, nor does it decisively prove the necessity of the proposed complexity over simpler baselines.

**Reviewer Scores:**

The reviewers would likely have kept their scores similar due to the persistent concern that the core contribution is an incremental modification of Sine-LoRA, and the theoretical concerns would result in a borderline rating.

---

### Decision · Program_Chairs · 2026-01-26

Reject